# Cytoplasmic FUS triggers early behavioral alterations linked to cortical neuronal hyperactivity and inhibitory synaptic defects

Jelena Scekic-Zahirovic[1,14], Inmaculada Sanjuan-Ruiz [1,14], Vanessa Kan[2,3], Salim Megat[1], Pierre De Rossi [4], Stéphane Dieterlé[1], Raphaelle Cassel[1,5], Marguerite Jamet[1], Pascal Kessler [6], Diana Wiesner[7,8], Laura Tzeplaeff[5], Valérie Demais[9], Sonu Sahadevan[4], Katharina M. Hembach [4], Hans-Peter Muller[7], Gina Picchiarelli[1], Nibha Mishra[10,11], Stefano Antonucci [7,8], Sylvie Dirrig-Grosch[1], Jan Kassubek [7], Volker Rasche [12], Albert Ludolph[7,8], Anne-Laurence Boutillier [5], Francesco Roselli [7,8], Magdalini Polymenidou [4], Clotilde Lagier-Tourenne [10,11], Sabine Liebscher [2,3,13,15✉] & Luc Dupuis [1,15✉]

Gene mutations causing cytoplasmic mislocalization of the RNA-binding protein FUS lead to severe forms of amyotrophic lateral sclerosis (ALS). Cytoplasmic accumulation of FUS is also observed in other diseases, with unknown consequences. Here, we show that cytoplasmic mislocalization of FUS drives behavioral abnormalities in knock-in mice, including locomotor hyperactivity and alterations in social interactions, in the absence of widespread neuronal loss. Mechanistically, we identified a progressive increase in neuronal activity in the frontal cortex of *Fus* knock-in mice in vivo, associated with altered synaptic gene expression. Synaptic ultrastructural and morphological defects were more pronounced in inhibitory than excitatory synapses and associated with increased synaptosomal levels of FUS and its RNA targets. Thus, cytoplasmic FUS triggers synaptic deficits, which is leading to increased neuronal activity in frontal cortex and causing related behavioral phenotypes. These results indicate that FUS mislocalization may trigger deleterious phenotypes beyond motor neuron impairment in ALS, likely relevant also for other neurodegenerative diseases characterized by FUS mislocalization.

[1] Université de Strasbourg, Inserm, Mécanismes centraux et périphériques de la neurodégénérescence, Strasbourg, France. [2] Institute of Clinical Neuroimmunology, Klinikum der Universität München, Ludwig-Maximilians-University Munich, Munich, Germany. [3] BioMedical Center, Medical Faculty, Ludwig-Maximilians-University Munich, Munich, Germany. [4] Department of Quantitative Biomedicine, University of Zurich, Zürich, Switzerland. [5] Université de Strasbourg, UMR 7364 CNRS, Laboratoire de Neurosciences Cognitives et Adaptatives (LNCA), Strasbourg, France. [6] Université de Strasbourg, Inserm, Unité mixte de service du CRBS, UMS 038, Strasbourg, France. [7] Department of Neurology, Ulm University, Ulm, Germany. [8] Deutsches Zentrum für Neurodegenerative Erkrankungen (DZNE), Ulm, Germany. [9] Plateforme Imagerie In Vitro, CNRS UPS-3156, NeuroPôle, Strasbourg, France. [10] Department of Neurology, The Sean M. Healey and AMG Center for ALS at Mass General, Massachusetts General Hospital, Harvard Medical School, Boston, MA, USA. [11] Broad Institute of Harvard University and MIT, Cambridge, MA, USA. [12] Ulm University Medical Center, Department of Internal Medicine II, Ulm, Germany. [13] Munich Cluster for Systems Neurology (SyNergy), Munich, Germany. [14] These authors contributed equally: Jelena Scekic-Zahirovic, Inmaculada Sanjuan-Ruiz. [15] These authors jointly supervised this work: Sabine Liebscher, Luc Dupuis. ✉email: sabine.liebscher@med.uni-muenchen.de; ldupuis@unistra.fr

Amyotrophic lateral sclerosis (ALS) is the major adult motor neuron disease, with onset usually in the 6th and 7th decade of life and death due to respiratory insufficiency and progressive paralysis typically occurring 3–5 years after onset of motor symptoms[1–3]. Mutations in the Fused in Sarcoma gene (FUS), encoding an RNA-binding protein from the FET family[4,5], are associated with the most severe forms of ALS[6,7], clinically presenting with a very early onset and rapid disease progression[8,9]. ALS associated mutations in FUS are clustered in the C-terminal region of the FUS protein that includes the atypical PY nuclear localization sequence, and is required for protein entry into the nucleus[6,7,10–12]. The severity of the disease correlates with the degree of impairment of FUS nuclear import[11,12], and the most severe cases of ALS known to date, are indeed caused by mutations leading to the complete truncation of the PY-NLS[8,9].

A number of clinical and pathological studies suggest that FUS mislocalization to the cytoplasm and subsequent aggregation could be relevant beyond the few ALS-FUS cases. First, FUS mutations, although rare in non-ALS cases, have been found in cases with frontotemporal dementia, either isolated[13,14] or as an initial presentation of ALS-FTD[15,16], as well as in patients with initial chorea[17], mental retardation[18], psychosis or dementia[19], and essential tremor[20]. In the absence of FUS mutations, FUS mislocalization[21], or aggregation[22,23] were found to be widespread in sporadic ALS. FUS pathology also defines a subset of cases with FTD (FTD-FUS) with prominent atrophy of the caudate putamen[24–26], concomitant pathology of other FET proteins, such as TAF15 and EWSR1[12,27–30] and frequent psychiatric symptoms[28]. FUS aggregates have also been observed in spinocerebellar ataxia and Huntington's disease[31,32]. While FUS mislocalization appears to be a common feature in neurodegenerative diseases, its pathological consequences have not been thoroughly studied beyond motor neuron degeneration.

Neurons with FUS pathology show decreased levels of FUS in the nucleus, that might compromise a number of processes dependent on proper FUS levels such as transcription and splicing regulation or DNA damage repair[4]. Interestingly, loss of FUS alters the splicing of multiple mRNAs relevant to neuronal function[33,34], such as MAPT, encoding the TAU protein, and alters the stability of mRNAs, encoding relevant synaptic proteins such as GluA1 and SynGAP1[35–39]. However, loss of nuclear FUS levels is very efficiently compensated for by autoregulatory mechanisms as well as by other FET proteins, and loss of nuclear FUS remains limited as opposed to loss of nuclear TDP-43, observed in TDP-43 pathology[40]. Indeed, heterozygous Fus knock-in mice, which carry one mutant allele leading to cytoplasmic and not nuclear localization of FUS, only show marginal loss of nuclear FUS due to compensatory overexpression[10,41]. Beyond nuclear loss of function, accumulation of cytoplasmic FUS was found to be a critical event in ALS-FUS in multiple studies in mouse models. For instance, cytoplasmic FUS is necessary to cause motor neuron degeneration in ALS-FUS[10,41–46] as heterozygous Fus knock-in mouse models develop mild, late onset muscle weakness and motor neuron degeneration, but not haploinsufficient Fus knockout mice[10,41,46]. To date, there are few studies investigating whether the accumulation of cytoplasmic FUS might lead to phenotypes beyond motor neuron degeneration. Interestingly, FUS is also found at synaptic and dendritic sites[38,47–51], and Sahadevan, Hembach et al.[52] identify synaptic mRNA targets for FUS that are critical for synaptic formation, function and maintenance.

Here, we show that a partial cytoplasmic mislocalization of FUS in heterozygous Fus knock-in mice is sufficient to drive a panel of behavioral abnormalities, including locomotor hyperactivity and alterations in social interactions, which preceded motor neuron degeneration. Behavioral deficits were accompanied by ventricle enlargement and atrophy of several subcortical structures in the absence of widespread neuronal loss in the cortex. Mechanistically, we could identify a progressive increase in neuronal activity in the frontal cortex of Fus knock-in mice in vivo. Furthermore, we observed a coordinated downregulation of multiple genes related to synaptic function in the frontal cortex throughout adulthood, which were confirmed by ultrastructural and morphological defects of synapses. These synaptic defects were more profound in inhibitory compared to excitatory synapses and accompanied by increased levels of FUS protein as well as of 3 of its RNA targets (Fus, Nrxn1, and Gabra1) in synaptosomes of heterozygous Fus knock-in mice. Thus, FUS cytoplasmic enrichment is sufficient to trigger synaptic deficits, leading to increased neuronal activity and behavioral phenotypes. These findings suggest that FUS mislocalization could trigger deleterious phenotypes beyond impaired motor function that could be relevant for both ALS-FUS but also for other neurodegenerative diseases based on FUS mislocalization.

## Results

**Spontaneous locomotor hyperactivity in *Fus*^ΔNLS/+ mice.** Since FUS mislocalization and aggregation are observed in patients with various neurodegenerative diseases, we hypothesized that partial FUS cytoplasmic mislocalization in *Fus*^ΔNLS/+ mice could be sufficient to cause a number of behavioral phenotypes. Two independent cohorts of mice were analyzed at 4 months of age, before the appearance of motor impairment[41] and 10 months of age. Evaluation of basal motor activity in a familiar environment showed significantly increased locomotor activity in *Fus*^ΔNLS/+ mice over the 3 consecutive days of observation (Fig. 1a, b). Interestingly, this hyperactivity was observed throughout the entire night in 4-months-old *Fus*^ΔNLS/+ mice (Fig. 1a), but only during late night hours in older *Fus*^ΔNLS/+ mice (Fig. 1b). In the open field, ambulatory distance, duration of ambulation, mean speed and preference for peripheral quadrants over central quadrants were similar in 10-months-old *Fus*^ΔNLS/+ mice and wild-type littermates, indicating the absence of hyperactivity in a novel environment (Supplementary Fig. 1a, b). To further study potential anxiety-related phenotypes in *Fus*^ΔNLS/+ mice, we used the dark/light box test, based on the preference of mice for dark compartments over illuminated places. In this test, 10-months-old *Fus*^ΔNLS/+ mice and *Fus*^+/+ mice showed a similar latency to enter, similar frequency of transitions and similar duration to explore illuminated compartment (Supplementary Fig. 1c). Thus, *Fus*^ΔNLS/+ mice are hyperactive, but do not show evidence of anxious behaviors, at least at the ages tested.

**Mildly compromised consolidation of spatial memory in *Fus*^ΔNLS/+ mice.** To explore the possibility that behavioral phenotypes of *Fus*^ΔNLS/+ mice included spatial memory defects, we performed the Morris water maze test. This task requires hippocampal function, at least during acquisition and memory formation, but relies on a proper cortico-hippocampal dialog for longer retention times or remote memory retrieval (Fig. 1c)[53]. At 10 months of age, *Fus*^ΔNLS/+ mice performed similarly well to their *Fus*^+/+ littermates regarding the distance travelled and latency to find the hidden platform over training days (Fig. 1d, e). We then performed a probe trial 18 days after the last training and observed that, although both genotypes searched significantly in the target quadrant compared to nontarget areas, *Fus*^ΔNLS/+ mice displayed a slightly decreased performance to retrieve memory at this timepoint (Fig. 1f). Furthermore, *Fus*^ΔNLS/+ mice lost their previous memory significantly faster than wild-type

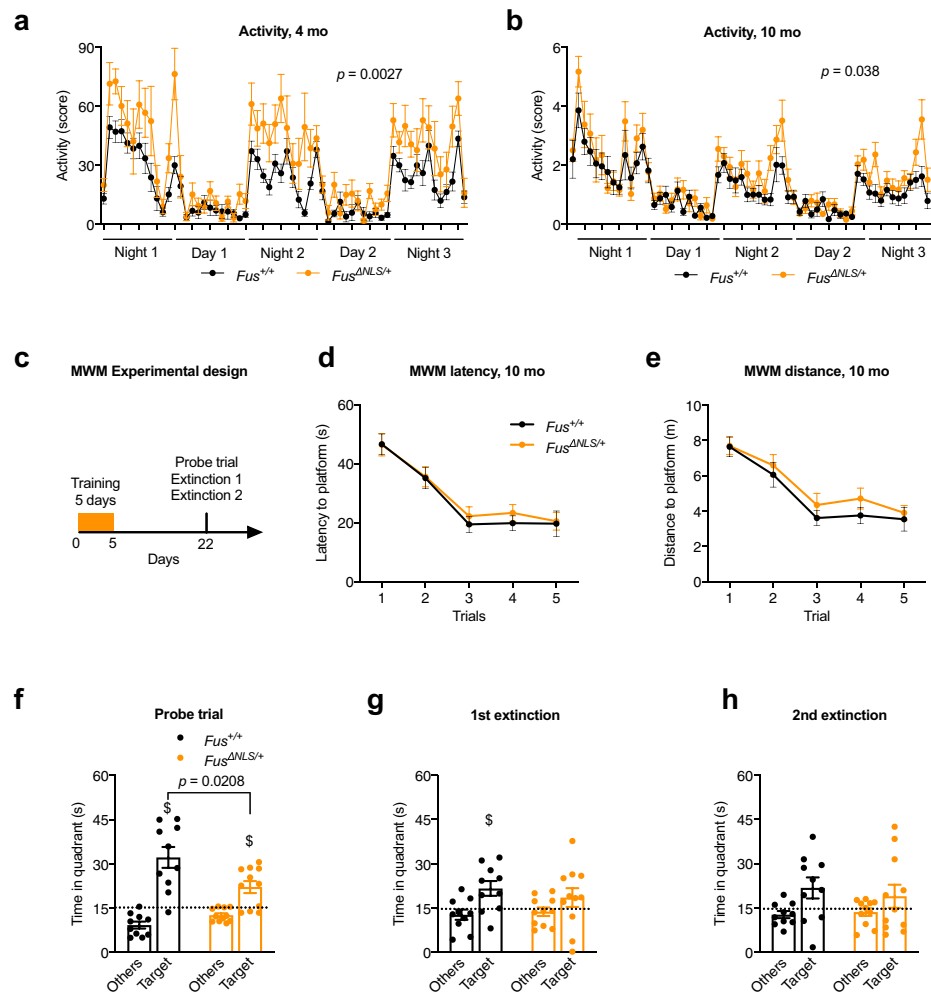

**Fig. 1 *Fus*ᐃNLS/+ mice display increased nocturnal spontaneous locomotor activity and cognitive defects. a, b** Line graphs represent mice home cage activity–actimetry over three consecutive days at 4 months (**a**) and 10 months (**b**) of *Fus*+/+ (black) and *Fus*ᐃNLS/+ (orange) male mice N = 11 for *Fus*+/+ and N = 10 for *Fus*ᐃNLS/+ mice at 4 months and N = 15 for *Fus*+/+ and N = 14 for *Fus*ᐃNLS/+ mice at 10 months. Repeated measures Two-way ANOVA followed by Sidak for multiple comparisons, with time and genotype as variables. P = 0.0027 at 4 months and p = 0.038 at 10 months for genotype effect. Data are presented as mean ± SEM values of activity score per hour. **c** Schematic illustration of the Morris water maze (MWM) experimental strategy (paradigm). Mice were subjected to a five-day training period and tested for spatial memory retention in a probe trial (60 seconds) 18 days after the last acquisition. The probe trial was then followed by two extinction tests, performed at 2 h intervals. **d, e** Line graphs represent latency (in seconds) (**d**) and total distance swam (in meters) (**e**) to find the hidden platform during acquisition of 10-months-old *Fus*+/+ (black) and *Fus*ᐃNLS/+ (orange) male mice. Both genotypes improved similarly their performance between day 1 and 5. N = 10 for *Fus*+/+ and N = 11 for *Fus*ᐃNLS/+ mice. Data are presented as mean ± SEM values of four trials per day of training. A two-way repeated measure analysis of variance (ANOVA) (genotype * days) was conducted to determine the effect of genotype on learning over time. No significant effect of genotype is observed. **f** Bar graphs represent the time spent in the target quadrant (Target) and the average of the time spent in the other three quadrants (Others) during probe trial. Dashed line indicates chance level (15 seconds per quadrant; i.e., 25%). N = 10 for *Fus*+/+ and N = 11 for *Fus*ᐃNLS/+ mice. Data are presented as mean ± SEM. Both genotypes were significantly above random but *Fus*ᐃNLS/+ mice performed significantly worse than *Fus*+/+ littermates ($, p < 0.01, One sample *t*-test was used to compare to a chance level, Target quadrant: p = 0.0008 for *Fus*+/+ and p = 0.006 for *Fus*ᐃNLS/+). Genotype comparison was made using One-way ANOVA; F(1,19) = 6.33, p = 0.0208. **g, h** Bar graphs represent the time spent in quadrants (Target vs Others) during the first (**g**) and the second (**h**) extinction test ($, p < 0.05 vs chance levels). One-way ANOVA for genotype effect (F(1,19) = 0.56, p = 0.46) (**g**), (F(1,19) = 0.27, p = 0.6) (**h**) and One sample *t*-test was used to compare to a chance level, (Target quadrant: p = 0.025 for *Fus*+/+ and p = 0.22 for *Fus*ᐃNLS/+) (**g**), (Target quadrant: p = 0.08 for *Fus*+/+ and p = 0.09 for *Fus*ᐃNLS/+) (**h**). N = 10 for *Fus*+/+ and N = 11 for *Fus*ᐃNLS/+ mice, with same mice as panel **f**. Data are presented as mean ± SEM. Source data are provided as a Source Data file.

mice, as they were searching randomly in a first extinction test performed 2 h after the probe trial, while wild-type mice still showed a significant more directed searching behaviour and preferred the target area over others (Fig. 1g). This suggests that consolidation of long-term memory was mildly compromised in *Fus*ᐃNLS/+ mice. Lastly, both genotypes did not distinguish the target over the other quadrants in a second extinction test (Fig. 1h). Altogether, these data show that *Fus*ᐃNLS/+ mice were

able to learn but displayed impaired long-term memory in agreement with a dysfunction of cortical regions.

**Social disinhibition in *Fus*ᐃNLS/+ mice.** Marked changes in personality and social behavior, such as social withdrawal or social disinhibition, obsessive-compulsive behaviors, euphoria or apathy are common in subjects with behavioral variant (bv)FTD, a disease with pronounced FUS mislocalization[54–56]. Social

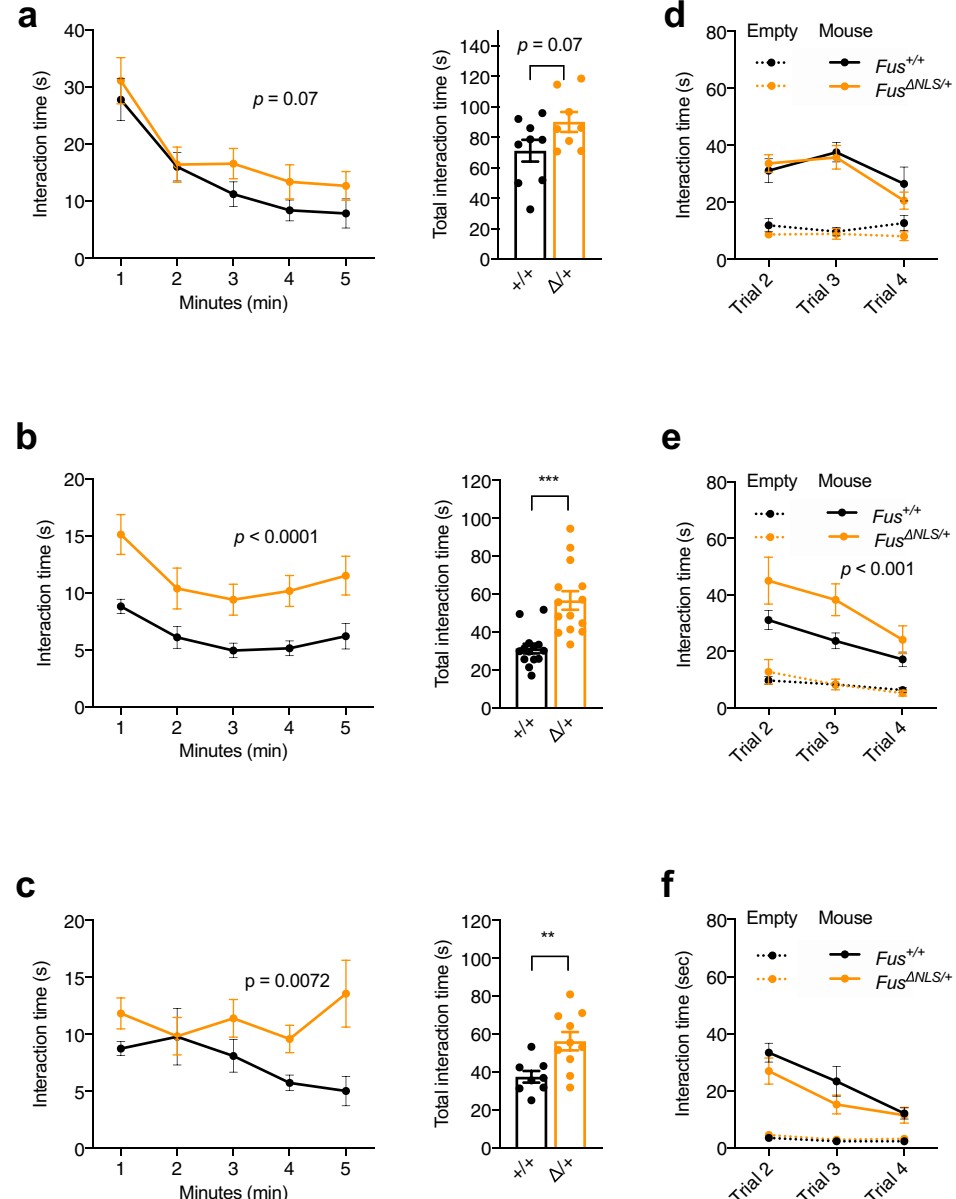

**Fig. 2 Social behavior abnormalities in $Fus^{\Delta NLS/+}$ mice. a–c** Line and bar graphs represent interaction time between resident (test) and intruder mice exclusively initiated by resident mouse in one-minute intervals (line graphs, on the left) or over the total time (bar graphs, on the right) during a 5 min resident-intruder test in home cage for 4 (**a**), 10 (**b**), and 22 (**c**) months-old $Fus^{+/+}$ (black) and $Fus^{\Delta NLS/+}$ (orange) male mice. Note that, young $Fus^{\Delta NLS/+}$ mice demonstrated a trend towards an increased social interest for intruder mouse (**a**) while older mice interacted with intruders significantly longer then $Fus^{+/+}$ (**b**, **c**) showing an age-dependent impairment of social behavior–disinhibition. All values are represented as mean ± SEM. At 4 months, $N = 9$ for $Fus^{+/+}$ and $N = 8$ for $Fus^{\Delta NLS/+}$ mice; At 10 months, $N = 14$ for $Fus^{+/+}$ and $N = 14$ for $Fus^{\Delta NLS/+}$ mice; At 22 months, $N = 8$ for $Fus^{+/+}$ and $N = 10$ for $Fus^{\Delta NLS/+}$ mice. Two-way repeated measures ANOVA followed by Sidak post-hoc test ($p = 0.07$ (4 months), $p < 0.001$ (10 months), and $p = 0.007$ (22 months) for genotype.effect); Two-sided Unpaired Student's $t$-test for total time $p = 0.07$ (4 months), $p < 0.001$ (10 months), and $p = 0.007$ (22 months)). **d**, **f** Line graphs represent sociability in the three-chamber test measured as interaction time with novel mice across three trials for $Fus^{+/+}$ (black) and $Fus^{\Delta NLS/+}$ (orange) male mice at 4 (**d**), 10 (**e**), and 22 (**f**) months of age. Time exploring an empty cage (object) across trials is represented as dashed lines. At 4 months, $N = 9$ for $Fus^{+/+}$ and $N = 8$ for $Fus^{\Delta NLS/+}$ mice; At 10 months, $N = 14$ for $Fus^{+/+}$ and $N = 14$ for $Fus^{\Delta NLS/+}$ mice; At 22 months, $N = 8$ for $Fus^{+/+}$ and $N = 9$ for $Fus^{\Delta NLS/+}$ mice. Data are presented as mean ± SEM. Three-way ANOVA with Newman Keuls post-hoc test for multiple comparisons, $p = ns$ (4 months), $p < 0.001$ (10 months) and $p = ns$ (22 months) for genotype effect). Source data are provided as a Source Data file.

deficits were also reported in progranulin haploinsufficient mice, a mouse model of FTD[57]. To determine whether $Fus^{\Delta NLS/+}$ mice have social behavioral deficits, we first performed the resident-intruder test specific for evaluating sociability in mice. Interestingly, 4-months-old $Fus^{\Delta NLS/+}$ mice showed a trend towards a

longer interaction with the intruder mouse as compared with $Fus^{+/+}$ mice ($p = 0.07$) (Fig. 2a), that was significant at 10 months of age (Fig. 2b) and persisted until 22 months of age (Fig. 2c). Aggressive behavior was only observed at 4 months of age, and not affected by the $Fus$ genotype (attack duration:

13.0 ± 1.4 s in $Fus^{+/+}$ mice vs 11.6 ± 1.0 s in $Fus^{\Delta NLS/+}$ mice, $p = 0.88$ two-sided unpaired Student's $t$-test). To further characterize the social behavioral impairment, we used a modified version of the three-chamber social paradigm. After a first trial of habituation using an empty setup, a novel mouse is introduced in a side compartment. The interactions initiated by the test mouse with either the novel mouse or the empty cage was quantified. Of most relevance, across the three consecutive trials (Trial 2, 3, and 4), we observed that 10-months-old $Fus^{\Delta NLS/+}$ mice consistently interacted more with the novel mouse than $Fus^{+/+}$ mice, in line with social disinhibition (Fig. 2e). This was not observed at 4 or 22 months of age (Fig. 2d–f). Importantly, mice of both genotypes spent more time interacting with the novel mouse than with the empty cage, indicating that mice could recognize its conspecific. The interaction time gradually decreased in later trials, suggesting progressive loss of social interest in the novel mouse, while it became familiar (Fig. 2d–f). Similar findings of social disinhibition in both resident-intruder test and three-chamber paradigms as a novel environment exclude the possibility that the observed increased social interactions resulted from locomotor hyperactivity in the home cage. Importantly, the olfactory function of $Fus^{\Delta NLS/+}$ mice was preserved, since results showed no differences between genotypes at 22 months of age in the time spent sniffing filter paper, covered with either attractive scent (vanilla) or an aversive scent (2-methyl butyrate) (Supplementary Fig. 2). These findings together with absence of major motor phenotype at that age (Supplementary Fig. 1a, b) indicated that social behavior is specifically affected in $Fus^{\Delta NLS/+}$ mice. Taken altogether, behavioral analyses of $Fus^{\Delta NLS/+}$ mice uncovered locomotor hyperactivity, cognitive deficits, and altered memory consolidation as well as selective impairment in sociability.

**Increased spontaneous neuronal activity in $Fus^{\Delta NLS/+}$ mice in vivo**. As the behavioral changes observed are highly reminiscent of frontal lobe dysfunction, we next asked whether neuronal activity is altered within that brain area. We thus examined spontaneous neuronal activity using in vivo two-photon calcium imaging (Fig. 3a–c). We studied neurons in cortical layer II/III of the frontal cortex expressing the genetically encoded calcium indicator GCaMP (delivered through an AAV vector) in mice at the age of 4 and 10 months (Fig. 3b). Indeed, we observed a significant increase in spontaneous activity, which worsened with age. While in 4-month-old mice the fraction of active neurons did not differ between $Fus^{+/+}$ and $Fus^{\Delta NLS/+}$ mice (Fig. 3d), there was a decrease in transient amplitudes (Fig. 3e) and an increase in transient frequency (Fig. 3f) in $Fus^{\Delta NLS/+}$ mice. In 10-month-old animals, this increase in activity was already evident at the level of the fraction of active cells in $Fus^{\Delta NLS/+}$ (Fig. 3g). Moreover, we observed an increase in the transient amplitudes (Fig. 3h) and also in the transient frequency (Fig. 3i) in $Fus^{\Delta NLS/+}$ mice compared to their $Fus^{+/+}$ littermates. Taken together, our data demonstrate an age dependent, strong increase in neuronal activity in vivo within the upper layers of frontal cortex of $Fus^{\Delta NLS/+}$ mice.

**$Fus^{\Delta NLS/+}$ mice show ventricle enlargement and atrophy of subcortical structures but preserved cortical neurons**. We next sought to understand the structural basis of behavioral and electrophysiological abnormalities in $Fus^{\Delta NLS/+}$ mice by employing MR imaging. FLASH MRI datasets for $Fus^{\Delta NLS/+}$ mice and $Fus^{+/+}$ littermates were processed for volumetric quantification using an in-house developed script[58], aimed at registering the MRI images to a template derived from the Allen Brain Atlas reference and then at parcellating the cerebral structures into hierarchically arranged volumes of interest, which can be interrogated for the volume of any region or group of regions (Fig. 4a, b). The overall intracranial volume (ICV) was comparable in $Fus^{\Delta NLS/+}$ and $Fus^{+/+}$ mice (Fig. 4c). However, upon normalization for the ICV, the volume of the brain parenchyma was significantly decreased in $Fus^{\Delta NLS/+}$ (by ~1.5%; average normalized volume was 98.52% for $Fus^{+/+}$ and 97.14% for $Fus^{\Delta NLS/+}$; Fig. 4d). Visual inspection of the MRI images revealed a substantial increase in the volume of lateral ventricles, which was confirmed by the registration algorithm and quantitated as an almost doubling of ventricular volumes (Fig. 4e). The ventriculomegaly was not associated with neocortical atrophy (Fig. 4f), but we identified a significant atrophy of the medial septum (Fig. 4g) and of the structures corresponding to the cortical subplate (including claustrum, endopiriform cortex and lateral, basomedial, basolateral, and posterior amygdalar nuclei; Fig. 4h). Only a nonsignificant trend for reduced volume was detected for hippocampus (9.06% for $Fus^{+/+}$ vs. 8.62% for $Fus^{\Delta NLS/+}$; $p = 0.15$; Two-sided Unpaired Student's $t$-test) and striatum (9.76% for $Fus^{+/+}$ vs. 9.89% for $Fus^{\Delta NLS/+}$; $p = 0.56$, Two-sided Unpaired Student's $t$-test). Interestingly, we also detected a significant degree of atrophy in the non-neocortical olfactory areas of the piriform cortex (2.48% for $Fus^{+/+}$ vs. 2.18% for $Fus^{\Delta NLS/+}$; $p = 0.0006$, Two-sided Unpaired Student's $t$-test). The lack of a prominent cortical atrophy phenotype was further confirmed by brain histology in $Fus^{\Delta NLS/+}$ mice at both 10 and 22 months of age. Cortical cytoarchitecture appeared preserved in $Fus^{\Delta NLS/+}$ mice, with normal lamination and no cortical thinning. The density of NeuN positive neurons in the frontal cortex was similar between $Fus^{\Delta NLS/+}$ mice and their wild-type littermates at 10 and 22 months of age (Fig. 4i, j).

Taken together, these data demonstrate a significant hydrocephalus ex vacuo in $Fus^{\Delta NLS/+}$, due to the atrophy of subcortical structures, such as the medial septum, several amygdalar nuclei, piriform areas, and tentatively the hippocampus.

**Transcriptome of $Fus^{\Delta NLS/+}$ cortex points to defects in inhibitory neurotransmission and synapses**. To understand the molecular basis of altered behavior in $Fus^{\Delta NLS/+}$ mice, we performed RNAseq on frontal cortex of 5- and 22-months-old $Fus^{\Delta NLS/+}$ mice and their wild-type littermates. Principal component analysis showed a clear separation between $Fus^{\Delta NLS/+}$ mice and their wild-type littermates at 22 months of age, while clustering was imperfect at 5 months of age, suggesting an exacerbation of the transcriptional differences between genotypes with age (Supplementary Fig. 3a).

Using a stringent analytical pipeline (FDR < 0.05), we did not identify differentially expressed genes between $Fus^{\Delta NLS/+}$ and $Fus^{+/+}$ mice at 5 and 22 months (Supplementary Fig. 3b). To ensure that the absence of differentially expressed genes was not due to the stringent calibration of $p$-values, we compared the 5-months and 22-months-old $Fus^{+/+}$ mice RNAseq datasets to probe age-related alterations. We were able to detect more than 2000 genes differentially expressed between 5- and 22-months-old wild-type mice, at a 5% false discovery rate, demonstrating that this approach can reliably detect changes in gene expression (Supplementary Fig. 3b).

To place gene expression changes in a systems-level framework, we performed weighted-gene coexpression network analysis (WGCNA) across all available $Fus^{\Delta NLS/+}$ and $Fus^{+/+}$ datasets, including 5 and 22-months RNAseq, as well as 1 and 6 months RNAseq datasets from Sahadevan et al.[52]. Potential batch effects were removed using a negative binomial regression model to estimate batch effects based on the count matrix[59] (Supplementary Fig. 3c) and allowed clustering between genotypes (Supplementary Fig. 3d). WGCNA analysis allowed us to

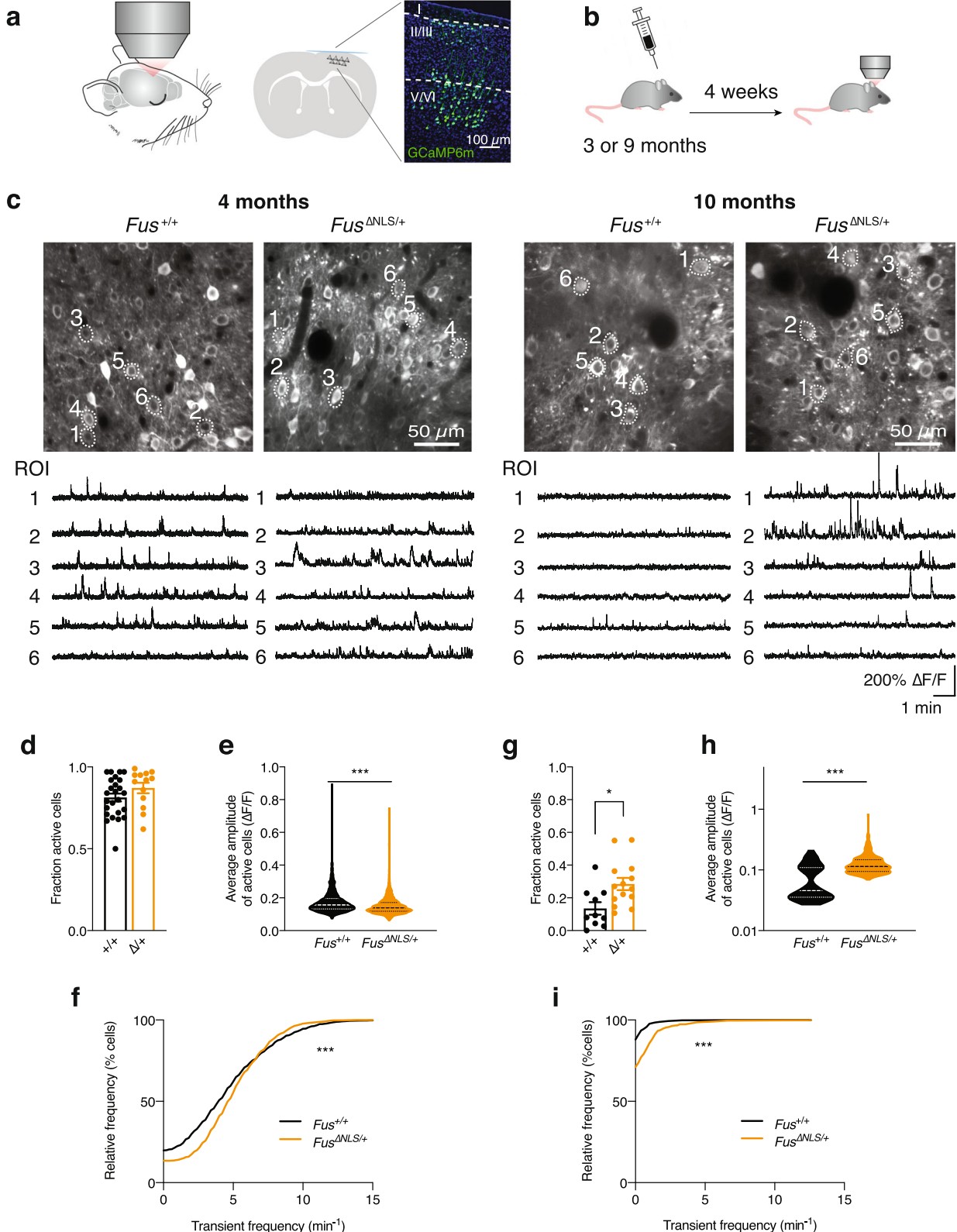

identify two mRNA modules significantly correlated with the genotype condition in cortex and labeled as turquoise and yellow modules according to the WGCNA conventions (Bonferroni-corrected $P < 0.05$; Fig. 5a, Supplementary Data 1). Cell-type enrichment analysis demonstrated that the turquoise module, but not the yellow module, was enriched in neuronally expressed

genes (Fig. 5b). Indeed, the Turquoise module, downregulated in $Fus^{\Delta NLS/+}$ mice (Fig. 5c–e), was enriched in genes related to synaptic physiology and development, most notably of GABAergic and glutamatergic synapses (Fig. 5d). Hub genes of the turquoise module included one GABA receptor encoding genes such as *Gabrb1*, one glutamate receptor gene (*Grid2*) and genes

**Fig. 3 Assessment of neuronal activity in $Fus^{\Delta NLS/+}$ mice in vivo. a** Neuronal activity was monitored in frontal cortex of anesthetized mice. Scheme of coronal section, indicating the expression of GCaMP6s in cortex assessed through a cranial window. Magnified view of imaged cortical area demonstrates neuronal expression of GCaMP (green) across all cortical layers. **b** Timeline of experiments. Male and female mice were injected with AAV9-syn-jGCaMP7s (at 3 months of age) or AAV2/1-hsyn-GCaMP6m (at 9 months of age) into frontal cortex and implanted with a cranial window. In vivo imaging began 4 weeks after implantation. **c** Representative examples (average projections) of field of views (FOV) imaged in $Fus^{+/+}$ and $Fus^{\Delta NLS/+}$ mice at 4 months ($N = 8$ $Fus^{+/+}$ mice and $N = 3$ $Fus^{\Delta NLS/+}$ mice, left) and at 10 months ($N = 5$ $Fus^{+/+}$ mice and $N = 6$ $Fus^{\Delta NLS/+}$ mice, right) are shown together with fluorescence calcium traces of selected regions of interest (ROIs). **d** The fraction of active cells per FOV was not affected in 4-month-old $Fus^{\Delta NLS/+}$ mice. $N = 13$ FOVs in 3 $Fus^{\Delta NLS/+}$ and $N = 25$ FOVs in 8 $Fus^{+/+}$ mice. Data are presented as mean ± SEM $p = 0.1627$, Two-sided Unpaired Student's $t$-test. **e**, **f** The calcium transient frequencies (**e**) were increased while the average transient amplitudes (**f**) were decreased in $Fus^{\Delta NLS/+}$ mice. $N = 1107$ ROIs in 3 $Fus^{\Delta NLS/+}$ and $N = 2264$ ROIs in 8 $Fus^{+/+}$, superimposed by the median (**e**). Kolmogorov–Smirnov test, ***$p < 0.0001$ for both panel **e** and **f**. **g–i** The fraction of active cells per FOV (**g**) as well as (**h**) the frequencies and (**i**) the average amplitudes of calcium transients of each ROI were increased in 10-month-old $Fus^{\Delta NLS/+}$ mice. Data are individual FOVs (**g**; $N = 14$ FOVs in 6 $Fus^{\Delta NLS/+}$ and $N = 10$ FOVs in 5 $Fus^{+/+}$ mice) or individual ROIs (**h**, **i**; $N = 855$ ROIs in 6 $Fus^{\Delta NLS/+}$ and $N = 631$ ROIs in 5 $Fus^{+/+}$ mice) superimposed by the mean ± SEM (**g**) or the median (**h**, **i**). panel **g:** Two-tailed Unpaired Student's $t$-test, *$p = 0.0126$; panel **h** and **i**: Kolmogorov–Smirnov test, ***$p < 0.0001$ for both panels. Source data are provided as a Source Data file.

tightly associated with synaptic development and autism (*Nrxn1, Lrfn5, Plcb1, Erc2, Frmpd4, Tanc2, Ctnnd2, Dmd*). Consistent with the known molecular function of FUS, the yellow module was enriched for genes related to RNA metabolism and processing and was progressively upregulated with age. Hub genes of this module comprise genes related to mRNA splicing (*Snrnp70, Ddx39b, Ilf3*), RNA transport (*Hnrnpl, Rbm3, Ipo4*), or RNA degradation (*Exosc10*) (Fig. 5f–h). Thus, transcriptome analysis points to the existence of synaptic defects in the frontal cortex of $Fus^{\Delta NLS/+}$ mice.

**Synaptic defects in in $Fus^{\Delta NLS/+}$ mice.** To independently validate potential synaptic defects in $Fus^{\Delta NLS/+}$ frontal cortex, we performed quantitative ultrastructural analysis of inhibitory (Fig. 6a) and excitatory (Fig. 6b) synapses in this brain region. Inhibitory synapses in layers II/III of the frontal cortex, identified by the presence of mitochondria on both sides of the synapse, showed major ultrastructural alterations in $Fus^{\Delta NLS/+}$ mice, with increased boutons sizes (Fig. 6c), longer active zones (Fig. 6d), prominently increased vesicle numbers (Fig. 6e), and increased distance of vesicles to the active zone as compared to wild-type synapses (Fig. 6f). Excitatory synapses, identified as asymmetrical, with a pronounced postsynaptic density, also showed ultrastructural alterations; however, in the opposite direction: excitatory synapses showed overall decreased bouton size, decreased length of the active zone, and decreased vesicle number in $Fus^{\Delta NLS/+}$ cortex (Fig. 6g–l). Importantly, ultrastructural alterations of excitatory synapses were less pronounced than those of inhibitory synapses.

To further explore morphological changes occurring at inhibitory synapses, we quantified the density and the cluster size of three inhibitory synaptic markers: the GABA transporter VGAT localized at the presynaptic site[60] and two receptors specifically expressed at the postsynaptic site of all GABA monoaminergic synapses[61], the postsynaptic scaffold protein Gephyrin[62] and the GABA$_A$ receptor containing α3 subunit (GABA$_A$Rα3). Pictures were acquired in cortical layer 1 to allow imaging of inhibitory synapses located on the apical dendrites of pyramidal neurons[63]. Consistent with the observed ultrastructural abnormalities, a significant decrease in all markers for inhibitory synapses was identified (Fig. 6k, l). This decrease in density was associated with a decrease in the size of the clusters for VGAT, GABA$_A$Rα3, and Gephyrin (Fig. 6m), suggesting a functional impairment of the remaining synapses.

We then sought to determine whether these defects in inhibitory synapses were caused or associated with the loss of inhibitory neurons and focused on parvalbumin-positive (PV) interneurons as the largest group of inhibitory interneurons in the cortex. Using immunohistochemistry, we did not detect differences in the number of PV neurons in the frontal cortex of $Fus^{\Delta NLS/+}$ mice neither at 10 nor at 22 months of age (Supplementary Fig. 4a–c). As a result of the ΔNLS mutation, FUS would be expected to accumulate in the cytoplasm of PV neurons as previously shown in other cell types[10,41,49]. We thus performed double immunostaining for FUS and parvalbumin and determined the nuclear/cytoplasmic ratio selectively in PV neurons. As shown in Supplementary Fig. 4d, e, cytoplasmic FUS staining was increased in PV neurons of $Fus^{\Delta NLS/+}$ compared to $Fus^{+/+}$ mice. Intriguingly, FUS cytoplasmic staining increased with age in wild-type PV interneurons, but remained significantly lower than in $Fus^{\Delta NLS/+}$ neurons. Altogether, these results demonstrate the existence of defects in cortical $Fus^{\Delta NLS/+}$ synapses, affecting inhibitory synapses more prominently, which could underlie the observed neuronal hyperexcitability (Fig. 3).

**Synaptic accumulation of FUS and its RNA targets in $Fus^{\Delta NLS/+}$ cortex.** To determine whether the observed phenotypes could be linked to a disrupted function of FUS at the synapse, we performed synaptosomal fractionation of the frontal cortex from 5-months-old $Fus^{\Delta NLS/+}$ mice. Obtained fractions were enriched in the synaptophysin protein (Fig. 7a–c, and Source data for uncropped western blots) and depleted in the nuclear lncRNA *Malat* (Fig. 7d), consistent with synaptic enrichment. In synaptosomes of $Fus^{\Delta NLS/+}$ mice, we observed an almost ten-fold increase in FUS content compared to wild-type synaptosomes, while the total or cytoplasmic FUS contents only increased 2–3 times (Fig. 7a–d). This increased FUS content was mostly due to mutant FUS synaptosomal accumulation, since it was not observed when using an antibody targeting the NLS of FUS (and thus not the mutant FUS ΔNLS protein) (Fig. 7a–d). FUS is known to bind a number of mRNAs, including *Fus* mRNA itself, as well as mRNAs important for (inhibitory) synaptic function such as *Nrxn1* or *Gabra1*[34]. Consistently, we observed increased levels of these 3 mRNAs in synaptosomal fractions of $Fus^{\Delta NLS/+}$ mice (Fig. 7e). This enrichment was relatively selective as 3 mRNAs encoding genes from the Turquoise module showed distinct patterns of synaptosomal enrichment: *Gabrb1* and *Grid2*, but not *Ctnnd2*, mRNAs showed clear synaptosomal enrichment, but only *Gabrb1* mRNA showed slightly elevated levels in $Fus^{\Delta NLS/+}$ synaptosomes. Collectively, our data show that defects in synapses, which are more pronounced in inhibitory synapses, and are related to synaptic FUS accumulation, likely causing the increased spontaneous neuronal activity and subsequent widespread behavioral abnormalities in $Fus^{\Delta NLS/+}$ mice.

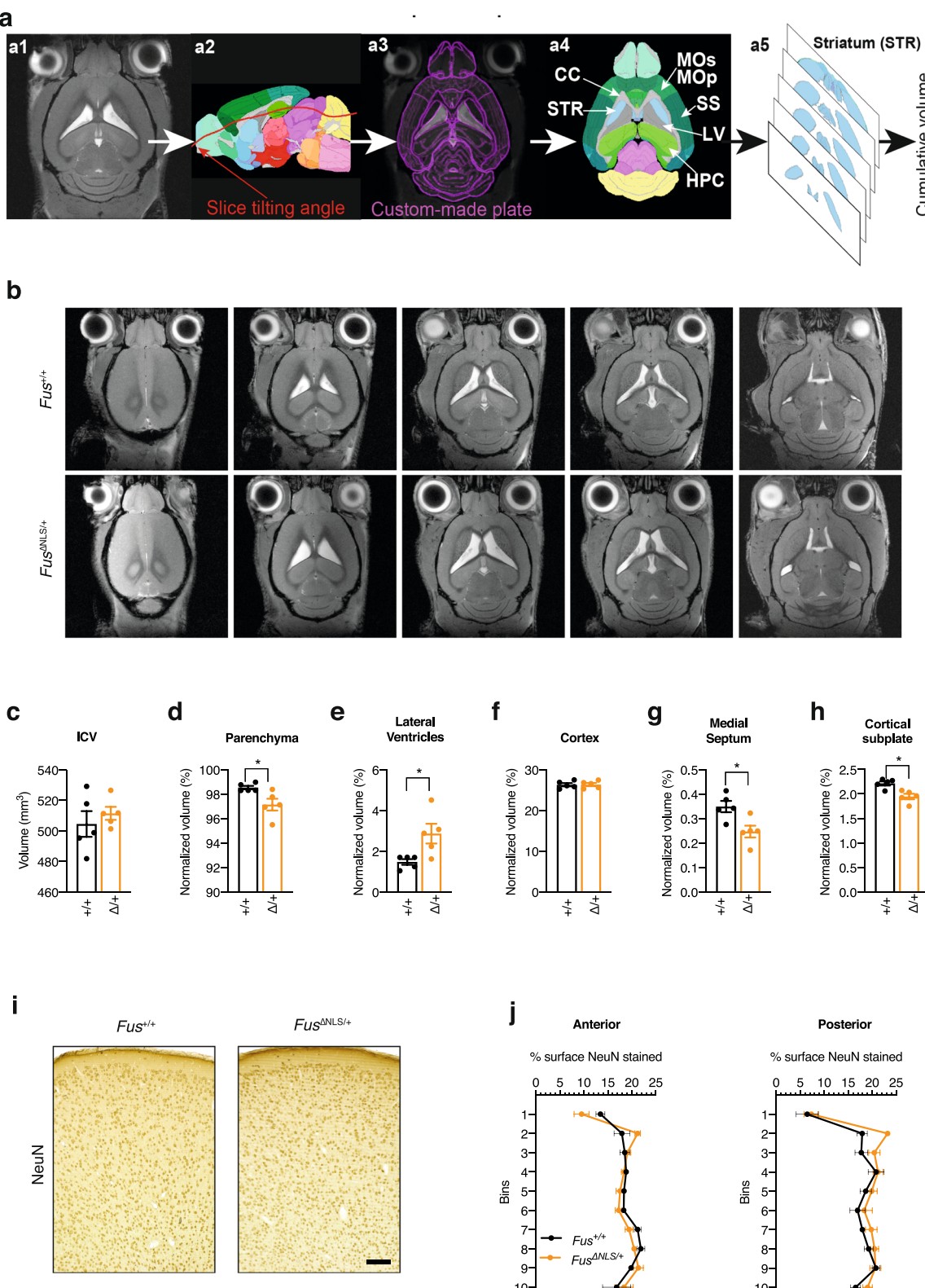

## Discussion

In this study, we show that knock-in mice with cytoplasmic accumulation of FUS display widespread behavioral alterations, beyond motor symptoms. We further determine that FUS mislocalization leads to increased spontaneous neuronal activity in the cortex, indicative of neuronal hyperexcitability, that is associated with structural and ultrastructural alterations of inhibitory synapses. Last, we show that the FUS mutation alters FUS synaptic content and modifies synaptic levels of a subset of its RNA targets, possibly underlying the observed phenotypes. The

**Fig. 4 Structural and histological brain analysis of $Fus^{+/+}$ and $Fus^{\Delta NLS/+}$ mice. a** Representation of the workflow used to determine volumes of corresponding brain structures from MRI slices per each mouse by the custom-made *Fiji* macro plugin (upper row). **b** Representative MRI slice images of $Fus^{+/+}$ (upper row) and $Fus^{\Delta NLS/+}$ (lower row) male mice. **c–h** Bar graph showing intracranial volume (ICV) (**c**), normalized volume of the brain parenchyma (**d**), of lateral ventricles (**e**), cortex (**f**), medial septum (**g**), and cortical subplate (**h**) in $Fus^{\Delta NLS/+}$ vs $Fus^{+/+}$ mice. For panels **c–h**, $N = 5$ for $Fus^{+/+}$ and $N = 5$ for $Fus^{\Delta NLS/+}$ mice. Data are presented as mean ± SEM. Two-tailed Unpaired Student's t-test, **c**: $p = 0.4838$; **d**: $p = 0.0249$; **e**: $p = 0.0249$; **f**: 0.9489; **g**: $p = 0.0151$; **h**: $p = 0.0051$. **i** Representative image of NeuN immunohistochemistry at 22 months of age in $Fus^{+/+}$ ($N = 3$ mice) or $Fus^{\Delta NLS/+}$ ($N = 5$) male mice in the anterior region of the M1/M2 cerebral cortex. Scale bar: 100 μm. **j** Distribution of NeuN+ neurons in $Fus^{+/+}$ (black) or $Fus^{\Delta NLS/+}$ (orange) male mice, in anterior and posterior regions of the M1/M2 cerebral cortex. $N = 3$ for $Fus^{+/+}$ and $N = 5$ for $Fus^{\Delta NLS/+}$ mice. Source data are provided as a Source Data file.

timelines of the different experimental studies are summarized in Supplementary Fig. 5.

The notion that FUS mislocalization is a widespread pathological event in sporadic ALS, but also in many other neurological diseases, prompted us to investigate the behavioral phenotype of $Fus^{\Delta NLS/+}$ mice. While motor defects can be detected as early as 6 months of age and motor neuron degeneration is not detected before 18–22 months of age, we observed an early spontaneous locomotor hyperactivity in $Fus^{\Delta NLS/+}$ mice. In addition, we observed various defects in executive functions, including impaired remote long-term memory, and abnormal social interactions. Hyperactivity and social and executive dysfunctions have been previously documented in other mouse models of ALS/FTD. As such the transgenic overexpression of mutant FUS can e.g., cause hyperactivity and cognitive deficits[64]. Similar abnormalities are also observed in TDP-43 knock-in mice[65], C9ORF72 BAC transgenic mice[66], or Chmp2b transgenic mice[67], suggesting that ALS mutations commonly lead to various behavioral alterations in mouse models, that are dominant over motor dysfunction. These phenotypes seen in mouse models nicely recapitulate widespread cognitive and executive dysfunction typical of ALS[68,69] and support the clinical overlap between ALS and FTD[70].

The deficits in executive functions and social behavior that we observe in $Fus^{\Delta NLS/+}$ mice are particularly relevant for FTD. Increased ventricular volume[71–73] as well as atrophy of subcortical structures[73,74] were found in FTD patients and presymptomatic mutation carriers, strengthening the analogy to $Fus^{\Delta NLS/+}$ mice. Pathology of FUS and other FET proteins (TAF15 and EWRS1) is a hallmark of a subset of FTD cases (FTD-FET cases). In FTD-FET cases, FUS pathology is associated with nuclear clearance of the FUS protein in neurons with FUS aggregates, although this nuclear clearance is not as pronounced as in cases with TDP-43 pathology[40]. Importantly, the FUS protein is accompanied by several other proteins in FTD-FET pathological aggregates, including TAF15 and EWSR1, two other FET proteins, as well as Transportin 1[12,27–30]. Thus, the disease in FTD-FET patients could be driven by several non-mutually exclusive mechanisms, including cytoplasmic accumulation and/ or aggregation of FUS, nuclear clearance of FUS and/or aggregation of co-deposited pathological proteins. Previous studies indicate that complete loss of FUS could be sufficient to lead to FTD like symptoms in mice, and this was consistent with the role of FUS in controlling the splicing of mRNAs relevant to FTD, such as *MAPT*, encoding the TAU protein, or in the stability of mRNAs encoding relevant synaptic proteins such as GluA1 and SynGAP1[35–39]. In $Fus^{\Delta NLS/+}$ mice, there is, however, a limited loss of nuclear FUS immunoreactivity[10,41] and no obvious FUS aggregates, ruling out that these pathological events might play a major role in the observed behavioral alterations. The quasi-normal levels of FUS in the nucleus are explained by the existence of potent autoregulatory mechanisms, which are able to largely buffer the effect of the mutation on nuclear FUS levels. Mislocalization of either TAF15 or EWSR1 is also unable to account

for behavioral abnormalities as both of these proteins show normal localization in $Fus^{\Delta NLS/+}$ neurons, as well as ALS-FUS patients[30]. Together, our results show that FUS mislocalization alone is sufficient to trigger behavioral symptoms and suggest that this might be a major driver of disease pathophysiology in FTD-FET patients. Importantly, our findings do not exclude that at later stages of disease progression, loss of nuclear FUS function might occur as a result of collapsed autoregulatory mechanisms, thereby exacerbating neurological symptoms.

A major finding of this study is that $Fus^{\Delta NLS/+}$ mice develop morphological and ultrastructural synaptic defects. The combination of locomotor hyperactivity with social deficits, as observed in $Fus^{\Delta NLS/+}$ mice, is commonly observed in various mouse models with synaptic defects. For instance, mouse models of autism spectrum disorders, such as mice lacking the ProSAP/ Shank proteins[75,76], display similar behavioral alterations. Our results point to a major defect in synapses, primarily affecting inhibitory synapses. This conclusion is supported by at least three main results: First, transcriptome analyses of the cerebral cortex show that genes related to synapses are affected. Second, the density of inhibitory synapses as well as the clusters size of three typical markers of inhibitory synapses (VGAT, GABA$_A$Ra3, and Gephyrin) are decreased. Third, inhibitory synapses are ultrastructurally abnormal, with increased size, increased number of vesicles and increased distance between vesicles and the active zone, which could be compensatory to their decreased density. Excitatory synapses were also abnormal, but their defects were minor compared to inhibitory synapses. Our data suggest that both the pre- and postsynaptic compartment of inhibitory synapses are affected by the *Fus* mutation. Indeed, the decrease in the density of Gephyrin positive puncta could reflect a disorganization of the postsynaptic density[77], potentially caused by decreased GABAR activity[78–80]. Decreased VGAT density, as well as increased bouton size or vesicle disorganization further suggest impairment of presynaptic GABAergic terminals. On its own, decreased VGAT density might reflect an overall reduction of inhibitory synapses throughout the cortical layers[81] and lead to impaired loading of GABA in the presynaptic vesicles[60]. Importantly, Sahadevan, Hembach and collaborators performed studies in $Fus^{\Delta NLS/+}$ mice at earlier ages and observed defects of inhibitory synapses, as early as 1 month of age, worsening at 6 months of age[52]. It is important to note that the disruption of inhibitory synapses can explain most of the detected behavioral and electrophysiological phenotypes observed in $Fus^{\Delta NLS/+}$ mice. Illustrating this, loss of *Gabra1*[82], or of *Gabra3*[83] are sufficient to lead to locomotor hyperactivity and the FUS target *Nrxn1* (encoding a key factor in the formation of GABAergic and glutamatergic synapses[84]), is critical in regulating locomotor activity and social behavior in mice[85,86]. Indeed, the deletion of all three neurexins from PV neurons is causing a decrease in the number of synapses of this neuronal type[87], in a manner similar to what is observed in $Fus^{\Delta NLS/+}$ mice.

Our current results do not allow to determine whether a specific subpopulation of inhibitory neurons would be more

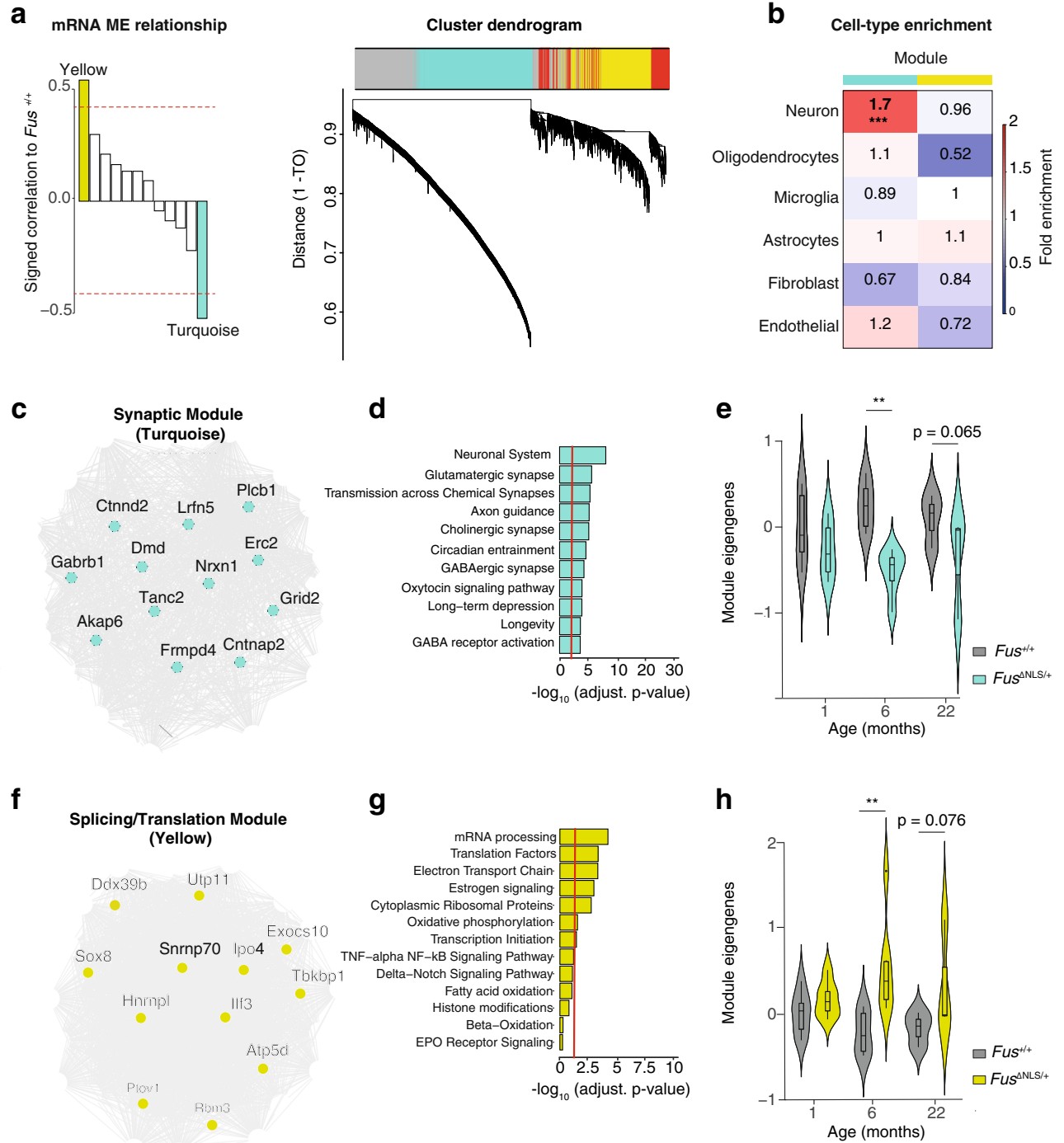

**Fig. 5 mRNA coexpression network analysis pinpoints defects in inhibitory and excitatory synapses in *Fus*$^{\Delta NLS/+}$ mice. a** Signed association (Pearson correlation) of the mRNA MEs with transgenic condition. Modules with positive values indicate increased expression in transgenic mice; modules with negative values indicate decreased expression in transgenic mice. The red dotted lines indicate Bonferroni-corrected $P < 0.05$ for multiple comparisons ($n = 12$ modules, $n = 16$ mice per group). **b** Cell-type enrichment of modules (average $n = 200$ genes) using mouse genes in mRNA modules (Fisher's two-tailed exact test, ***FDR = $2 \times 10^{-5}$). **c** Coexpression network plot of the synaptic (turquoise) module. The top 12 hub genes are indicated by name. **d** Gene ontology term enrichment of the synaptic module using 1791 synaptic module genes. **e** Trajectory of the synaptic module in the cortex of *Fus*$^{\Delta NLS/+}$ mice across time. Boxplot show median and quartile distributions, the upper and lower lines representing the 75th and 25th percentiles, respectively. Two-way ANOVA, $F_{(1,24)} = 14.55$, $p = 0.0008$; $n = 4$–6 mice per group. **f** Coexpression network plot of the splicing/translation module. The top 12 hub genes are indicated by gene name. **g** GO term enrichment of the splicing/translation module using 1112 splicing/translation module genes. **h** Trajectory of the splicing/translation module in the cortex of *Fus*$^{\Delta NLS/+}$ mice across time. Boxplot show median and quartile distributions, the upper and lower lines representing the 75th and 25th percentiles, respectively. Two-way ANOVA, $F_{(1,24)} = 11.92$, $p = 0.002$; $n = 4$–6 mice per group. The center line represents the median.

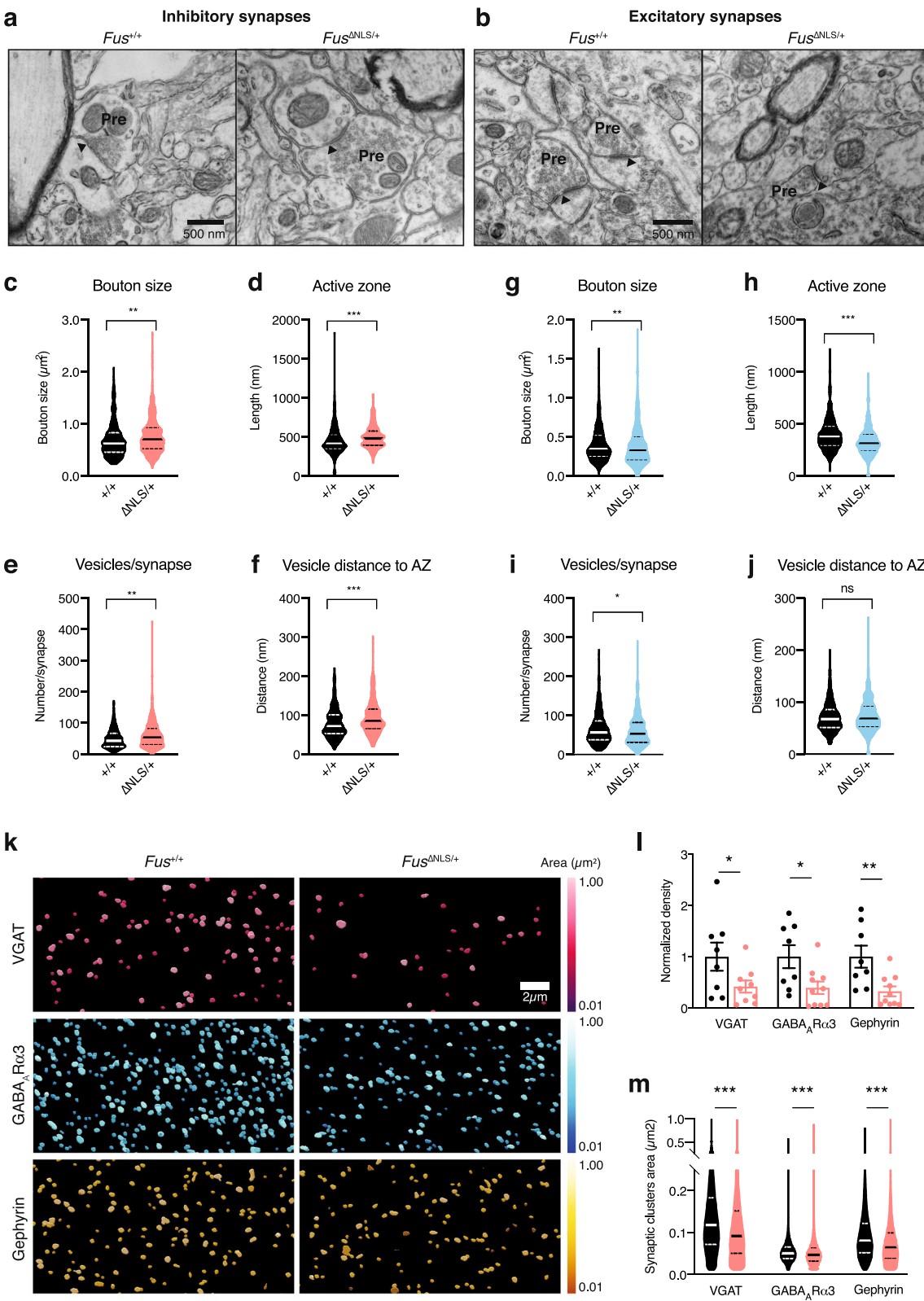

selectively affected in $Fus^{\Delta NLS/+}$ mice. PV interneurons are, however, a strong candidate according to the results of our studies, but also their involvement in TDP-43 knock-in mice[65], and in TDP-43 transgenic mice that display degeneration of hippocampal PV positive interneurons[88]. Functional impairment of PV interneurons might represent a unifying theme in ALS pathophysiology, as multiple electrophysiological studies demonstrate

hypoexcitability of PV neurons in SOD1 and TDP-43 transgenic mouse models of ALS[89–92]. Others, however, found PV interneurons to be unaltered presymptomatically and to turn hyperexcitable during the symptomatic phase in the same SOD1[G93A] mouse model[93]. In either case, those changes in PV excitability were always accompanied by hyperexcitability of layer V pyramidal neurons[89–91,93]. These findings in mouse models nicely

**Fig. 6 Defects in synapses in 22-months-old $Fus^{\Delta NLS/+}$ mice. a, b** Representative image of transmission electron microscopy in $Fus^{+/+}$ or $Fus^{\Delta NLS/+}$ layer II/III of the motor cortex at 22 months of age showing inhibitory synapses (**a**) (as containing ≥1 mitochondrion on each side of the synapse) and excitatory synapses (**b**). Pre: presynaptic compartment; active zone is shown with an arrowhead. $N = 4$ $Fus^{+/+}$ mice (1 male and 3 females), and $N = 4$ $Fus^{\Delta NLS/+}$ mice (1 male and 3 females) have been analyzed. **c–f** Violin plot showing the distribution of bouton sizes (**c**), the length of active zones (**d**), the number of vesicles per synapse (**e**), and the distance of individual vesicles to the active zone (**f**) in inhibitory synapses of $Fus^{+/+}$ (black) or $Fus^{\Delta NLS/+}$ (orange) mice. For panels **c–f**, $N = 379$ synapses from 1 male and 3 female $Fus^{+/+}$ mice and $N = 387$ synapses from 1 male and 3 female $Fus^{\Delta NLS/+}$ mice were analyzed. Kolmogorov–Smirnov test. **c**: $p = 0.0016$; **d**: $p < 0.0001$; **e**: $p = 0.0010$; **f**: $p < 0.0001$. **g–j** Violin plot showing the distribution of bouton size (**g**), the length of active zone (**h**), the number of vesicles per synapse (**i**), and the distance of individual vesicles to the active zone (**j**) in excitatory synapses of $Fus^{+/+}$ (black) or $Fus^{\Delta NLS/+}$ (cyan) mice. For panels **g–j**, $N = 463$ synapses from 1 male and 3 female $Fus^{+/+}$ mice and $N = 490$ synapses from 1 male and 3 female $Fus^{\Delta NLS/+}$ mice were analyzed. Kolmogorov–Smirnov test. **g**: $p = 0.0038$; **h**: $p < 0.0001$; **i**: $p = 0.0362$; **j**: $p = 0.2182$. **k** Representative images of GABAARα3, Gephyrin and VGAT intensity in 22-months male mice, coded by area size (Imaris). $N = 3$ $Fus^{+/+}$ mice and $N = 4$ $Fus^{\Delta NLS/+}$ mice have been analyzed. **l** Bar graphs representing the density analysis for VGAT, GABAARα3, and Gephyrin comparing $Fus^{+/+}$ vs $Fus^{\Delta NLS/+}$ mice. ($Fus^{+/+}$ vs $Fus^{\Delta NLS/+}$, Mann–Whitney test, VGAT, $p = 0.0464$; GABAARα3, $p = 0.0217$; Gephyrin, $p = 0.0043$). $N = 8$ FOVs from 3 $Fus^{+/+}$ mice and $N = 9$ FOVs from 4 $Fus^{\Delta NLS/+}$ mice were analyzed for VGAT; $N = 8$ FOVs from 3 $Fus^{+/+}$ mice and $N = 10$ FOVs from 4 $Fus^{\Delta NLS/+}$ mice were analyzed for GABAARα3 and Gephyrin. Data are presented as mean ± SEM. Mann–Whitney, One tailed, VGAT: $p = 0.0464$; GABAARα3: $p = 0.0217$; Gephyrin: $p = 0.0043$. **m** Violin plot representing the analysis of the clusters size for VGAT, GABAARα3, and Gephyrin comparing $Fus^{+/+}$ vs $Fus^{\Delta NLS/+}$ mice. $N = 142,416$ synapses from 3 $Fus^{+/+}$ mice and $N = 115,151$ synapses from 4 $Fus^{\Delta NLS/+}$ mice were analyzed for VGAT; $N = 202,302$ synapses from 3 $Fus^{+/+}$ mice and $N = 99,464$ synapses from 4 $Fus^{\Delta NLS/+}$ mice were analyzed for GABAARα3; $N = 169,036$ synapses from 3 $Fus^{+/+}$ mice and $N = 68,422$ synapses from 4 $Fus^{\Delta NLS/+}$ mice were analyzed for Gephyrin. Kolmogorov–Smirnov test. VGAT: $p < 0.0001$; GABAARα3: $p < 0.0001$; Gephyrin: $p < 0.0001$. Source data are provided as a Source Data file.

recapitulate human ALS pathology, in which cortical hyper-excitability is a frequent and, most importantly, early finding in familial and sporadic cases, including FUS mutation carriers[16,94]. In line with these findings, we also observed a pronounced increase in spontaneous neuronal activity in vivo, which is highly indicative of hyperexcitable pyramidal neurons. While we cannot rule out cell autonomous alterations affecting the intrinsic excitability of pyramidal neurons, our histological, ultrastructural, and transcriptomic data strongly argue for defective inhibitory neurotransmission by PV interneurons. In summary, our results, along with others, support the notion that dysfunction of cortical PV interneurons contribute to neural circuit defects in ALS and FTD. Importantly, while we observe molecular and structural defects in inhibitory neurons, we did not observe a loss of PV cell bodies in $Fus^{\Delta NLS/+}$ mice, suggesting that the major defect resembles a synaptopathy rather than frank neuronal loss, consistent with other studies[51]. Altogether, our results identify a role for FUS in regulating GABAergic synapse structure and function. Since other major classes of inhibitory interneurons[95] were not investigated, we cannot exclude that somatostatin positive (SST) or HTR3A expressing interneurons are also affected, although to a lesser extent than PV neurons. Furthermore, our work also shows that this *Fus* mutation alters glutamatergic synapses, as judged from both WGCNA analysis of RNAseq (Fig. 5) and electron microscopy (Fig. 7). This is consistent with results from Sahadevan, Hembach et al.[52] providing evidence that FUS is also critically involved in glutamatergic synaptogenesis, at least during development, and is in line with previous studies[96]. Further work is required to disentangle the causes and consequences of GABAergic and glutamatergic impairment, and their respective mechanisms.

How can mutant FUS regulate inhibitory synaptic structure? We observe that the loss of the FUS NLS leads to an increased level of the mutant protein in purified synaptosomes. These results are consistent with results from Sahadevan, Hembach et al., where the authors identified a number of FUS synaptic RNA targets, and a subset of these were also increased in synaptosomes of $Fus^{\Delta NLS/+}$ mice. Interestingly, in both studies, several FUS synaptic targets are not modified in $Fus^{\Delta NLS/+}$ synaptosomes, including some related to GABAergic neurons. Sahadevan, Hembach and collaborators further demonstrate that at least a subset of these FUS synaptic RNA targets show

increased stability in $Fus^{\Delta NLS/+}$ neurons. It seems thus reasonable to hypothesize that accumulation of synaptic FUS compromises synaptic homeostasis through altered stability of key synaptic RNAs, either through direct binding, or indirectly. This does not exclude additional mechanisms of toxicity for synaptic FUS, in particular effects on local synaptic translation[44,97], that could affect synaptic protein levels. Further work should focus on determining whether FUS might also regulate synaptic translation of specific proteins involved in inhibitory transmission, and whether rescuing synaptic defects in inhibitory neurons might translate into an efficient therapeutic strategy.

In summary, we show here that cytoplasmic accumulation of FUS leads to a major synaptopathy mainly in inhibitory neurons, that is accompanied by consistent behavioral and electro-physiological phenotypes. The identification of the mechanisms downstream of FUS' synaptic action might lead to efficient therapeutic strategies for FUS related neurodegenerative diseases.

## Methods

**Mouse models and behavioral analyses.** Wild-type ($Fus^{+/+}$) and heterozygous ($Fus^{\Delta NLS/+}$) mice on a pure genetic background (C57BL/6 J), have been described previously[10], were bred and housed in the central animal facility of the Faculty of medicine of Strasbourg, with a regular 12-h light and dark cycle (light on at 7:00 am) under constant conditions (21 ± 1 °C; 60% humidity). Standard laboratory rodent food and water were available ad libitum throughout all experiments. Mice were genotyped by PCR of genomic DNA from tail biopsies using oligonucleotide primers (sequence provided in Supplementary Data 2)[10]. Mouse experiments were performed in compliance with all relevant ethical regulations for animal testing and researcher. All experiments were approved by local ethical committee from Strasbourg University (CREMEAS) under reference number AL/27/34/02/13 (behavior), by the Government of upper Bavaria (license number Az 55.2-1-54-2532-11-2016, two-photon microscopy) and by „Regierungspräsidium Tübingen" (animal license number 1431, MRI). Behavioral tests were done during the light phase (between 9 am and 5 pm) of their light/dark cycle except for indicated experiments. Until the mice reached the age when the behavioral tests were performed mice were group-housed. Once mice were single housed for the behavioral task they were kept individually for only a period necessary to finalize the set of behavioral experiments and in order to minimize possible negative effects of isolation, afterwards cohorts were sacrificed and processes for downstream analyses. Male mice of 4, 10, and 22 months of age were subjected to behavioral studies and data were analyzed blind to genotypes. The sex of the animals studied is indicated in each figure legend.

*Spontaneous locomotor activity in the home cage–actimetry.* Home cage activity was assessed according to previously published protocols[98]. Mice were placed individually in large transparent Makrolon cages (42 × 26 × 15 cm) adapted to the shelves of the testing device (eight cages/shelve). Two infrared light beams, passing through each cage, were targeted on two photocells, 2.5 cm above the cage floor

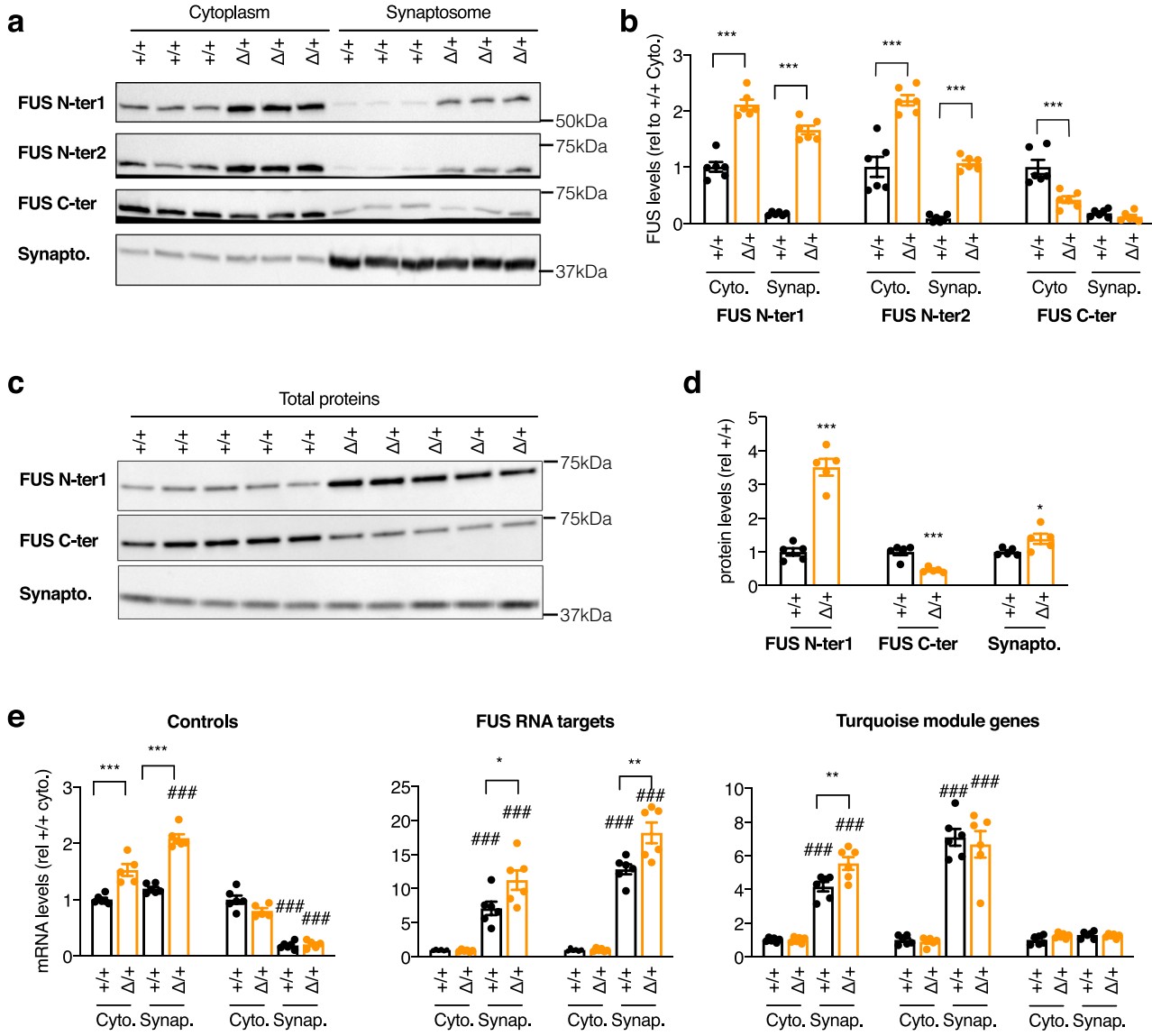

**Fig. 7 FUS accumulates in synaptosomes of *Fus*^ΔNLS/+ mice and alters synaptosomal levels of a subset of its targets. a, b** Representative western blot images (**a**) and respective quantifications (**b**) of cytoplasmic (**a**, left) or synaptosome (**a**, right) extracts from *Fus*^+/+ (+/+) or *Fus*^ΔNLS/+ (Δ/+) mice (4 months of age,) using two antibodies recognizing the N-terminal part of the FUS protein (FUS N-ter1 and FUS N-ter2), the C-terminal part of FUS (encoding the NLS, FUS C-ter) or synaptophysin protein to show enrichment in synaptic proteins in the synaptosome fraction. $N = 6$ *Fus*^+/+ mice and $N = 6$ *Fus*^ΔNLS/+ mice were analyzed. Data are presented as mean ± SEM. One-Way ANOVA with Tukey post-hoc test. ***$p < 0.0001$ Please note that the FUS western blots were run on independent gels, to avoid stripping and reprobing on the same membrane for the same protein. Each of these gels were controlled for equal loading using StainFree markers, that are provided in the source data. **c, d** Representative western blot images (**c**) and respective quantifications (**d**) of total extracts (**c**) from *Fus*^+/+ (+/+) or *Fus*^ΔNLS/+ (Δ/+) mice (4 months of age,) using the same antibodies as in panel **a**. $N = 5$ *Fus*^+/+ mice and $N = 5$ *Fus*^ΔNLS/+ mice were analyzed. Data are presented as mean ± SEM. Two-tailed Unpaired Student's *t*-test. N-ter1: $p < 0.0001$; C-ter: $p$-value: $p = 0.0006$; Synaptophysin: $p = 0.0411$. **e** mRNA levels of the indicated genes in RNAs extracted from cytoplasmic (Cyto.) or synaptosome (Synap.) extracts from *Fus*^+/+ (+/+) or *Fus*^ΔNLS/+ (Δ/+) frontal cortex from 4-months-old female mice as assessed using RT-qPCR. $N = 6$ *Fus*^+/+ mice and $N = 5$ *Fus*^ΔNLS/+ mice were analyzed. Data are presented as mean ± SEM. Genes are grouped by categories (controls, established FUS RNA targets, and genes belonging to the Turquoise module). All quantifications are presented relative to the +/+ cytoplasmic RNA levels set to 1. One-way ANOVA with Tukey post-hoc test. *Fus:* ***$p < 0.0001$ vs corresponding wild-type fraction; ###$p < 0.0001$ vs corresponding cytoplasmic fraction of the same genotype. *Malat:* ###$p = 0.0001$ vs corresponding cytoplasmic fraction of the same genotype. *Nrxn1* *$p = 0.0140$ vs corresponding wild-type fraction; ###$p < 0.0001$ vs corresponding cytoplasmic fraction of the same genotype. *Gabra1:* **$p = 0.0012$ vs corresponding wild-type fraction; ###$p < 0.0001$ vs corresponding cytoplasmic fraction of the same genotype. *Gabrb1:* **$p = 0.0029$ vs corresponding wild-type fraction; ###$p < 0.0001$ vs corresponding cytoplasmic fraction of the same genotype. *Grid2:* ###$p < 0.0001$ vs corresponding cytoplasmic fraction of the same genotype. *Ctnnd2:* no significant differences observed ($p > 0.05$). Source data are provided as a Source Data file.

level and 28 cm apart. The number of cage crossing was recorded automatically and was used to determine or score the spontaneous locomotor activity. The experiment began at 17.00 pm and after 2 h of habituation continued for 3 consecutive days for a complete 24 h nictemeral cycle (12 h dark and 12 h light).

*Open field*. The general exploratory locomotion and anxiety in a novel environment were tested during 15 min long sessions in the open field arena (72 × 72 × 36 cm) located in a test room and lit by a 600 lux for background lighting, according to published protocol[99]. The open maze was divided by lines into sixteen squares (18 × 18 cm). Each mouse was placed in the center of the arena and allowed to freely move while being video recorded. The recorded data were analyzed offline with EthoVision XT software system (Noldus Information Technology). The time spent in the center (four central quadrants) vs. the perimeter (12 peripheral quadrants) was used to measure anxiety, while the total distance traversed in the arena and average moving speed (mean velocity) was used to evaluate locomotor activity. For each mouse a movement heat map and trajectory tracking map that are representing a corresponding locomotor activity were made independently.

*Dark/light box test*. The light/dark box apparatus consisted of two Poly-Vinyl-Chloride (PVC) compartments of equal size (18.5 × 18.5 × 15 cm) one opaque and the other transparent, connected through an opaque tunnel (5 × 5.5 × 5 cm). The illumination of the transparent compartment was set at 400 lux. Each mouse was placed alone in the dark compartment and the mouse's behavior was recorded during 5 min with a video camcorder located ~150 cm above the center of the box. Test was conducted during the morning. The latency before the first transition into the light compartment, the number of transitions between the two compartments and the time spent in each compartment were tested to assess for anxiety level and exploratory behavior, as published previously[99].

*Olfactory preference test*. This test is designed to identify specific detection deficiencies and/or odor preference, namely the ability to sense attractive or aversive scents. After 30 min of habituation to empty cage with no bedding, each mouse was challenged with a filter paper embedded with two strong scents (vanilla and 2-methyl butyrate) or a neutral scent (water) was video recorded over 3 min. A 1-h pause in between the exposure to different scents was applied to each mouse using a procedure adapted from previously reported protocols[67,100]. The time the mouse spent sniffing the filter paper—the exploration time, is calculated post-hoc by an examiner blind to mouse genotype and condition. Those scents with the exploration time greater than water were designated as "attractive" while those with times less than water were termed "aversive".

*Social interaction in the home cage (resident-intruder test)*. Social interaction was assessed in the home cage by a standard protocol[67,101]. Briefly, both resident and intruder mice were isolated and housed individually for 1 week before the task. After 30 min of habituation to the test room resident mouse was allowed to freely roam in his home cage without the cage top for 1 min. A novel male intruder mouse (nonlittermate of same background, same age, and similar weight) was then introduced in the opposite corner as the resident and allowed to interact for 5 min while videotaped. The total physical interaction, defined as the time during which the resident mouse actively explores the intruder was analyzed post-hoc. Only social activities, such as time spent investigating, grooming, following, sniffing etc. were quantified separately for each minute and for the whole time of the task and were differentiated from nonsocial/aggressive activities such as attacks, bites, and tail rattles.

*Three-chamber social task*. Specific social behaviors such as sociability and social recognition were analyzed by using three-chamber social task. The experimental procedure is adapted from Gascon E et al.[67]. The three-chamber box (59 × 39.5 × 21.5 cm) is made of transparent Plexiglas (Noldus Information Technology, Wageningen, The Netherlands) and is divided into three chambers (one middle and two side chambers) of equal size (18.5 × 39.5 cm) by the walls with a square opening (7 × 7 cm) that could be closed by a slide door. Each of the two side chambers contains a mobile wire cylinder shaped cage (20 × 10 cm diameter) that is made of transparent Plexiglas bars placed 6 mm apart. Cage is closed by the upper and lower lids.

Mice of both genotypes (*Fus*[+/+] and *Fus*[ΔNLS/+]) that were experimentally tested are referred to as the test mice and adult male unfamiliar mice of same background, age and weight used as the social stimulus are called novel mice. All mice were housed individually for 1 week before the test and were habituated to the testing room for at least 1 h before the start of behavioral tasks. One day prior to the testing, the novel mice were habituated to mobile wire cage for 5 min. The keeping of the novel mouse separated in a wire cage prevents aggressive and sexual activities, and in the same time ensures that any social interaction is initiated by the test mouse. Sessions were videotaped and visually analyzed post-hoc. The experimental procedure was carried out in four trials of 5 min each. After each trial, the mouse was returned to his home cage for 15 min. Trials were grouped into two consecutive parts.

Trial 1 (habituation): the test mouse was placed in the middle chamber and left to freely explore each of the three chambers: the empty middle or two sides' arenas containing the empty wire cages for 5 min.

Trials 2–4 (sociability, social recognition, social learning acquisition): the mouse was placed in the middle chamber, but an unfamiliar mouse (novel mouse) was placed into a wire cage in one of the side chambers (the wire cage in the other side-chamber remains empty). The test mouse had free access to all three chambers. The position of novel mouse and empty wired cage were alternated between trials. We quantified the time spent actively exploring a novel mouse or an empty cage by the test mouse as a social interaction time or an object exploration time, respectively. The longer time that test mouse spends in the close perimeter around the cage containing the novel mouse while actively interacting with it (staring, sniffing) compared to the empty cage—object, indicates social preference or social recognition as a result of the capability to differentiate a conspecific from an object. The motivation of the test mouse to spontaneously interact with novel mouse is considered as sociability which gradually decreased over trials as a result of social learning acquisition.

*Water maze task*. The water maze consisted of a circular pool (diameter 160 cm; height 60 cm) filled with water (21 ± 1 °C) made opaque by addition of a powdered milk (about 1.5 g/L). The habituation day consisted in one 4-trial session using a visible platform (diameter 11 cm, painted black, protruding 1 cm above the water surface and located in the South-East quadrant of the pool), starting randomly from each of the four cardinal points at the edge of the pool. During this habituation trial, a blue curtain surrounded the pool to prevent the use of distal cues and thus incidental encoding of spatial information. For the following days, the curtain was removed. Mice were given a 5-day training period (4 consecutive trials/day, maximum duration of a trial 60 seconds, inter-trial interval = 10–15 seconds) with a hidden platform located at a fixed position in the North-West quadrant. Animals were starting randomly from each of the four cardinal points at the edge of the pool and the sequence of the start points was randomized over days. Mice were tested for retention in a 18-days delay probe trial and two extinction tests: the first 2 h after probe trial and the second 2 h after the first. For the probe trial, the platform was removed; the mice were introduced in the pool from the North-East (a starting point never used during acquisition) and allowed a 60-seconds swimming time to explore the pool. Data were collected and computed by a video-tracking system (SMART; AnyMaze software). For the visible platform and training trials following parameters were used: the distance traveled and the latency time before reaching the platform and the average swimming speed. For the probe trial and extinction tests the time (in secondes) spent in the target quadrant (i.e., where the platform was located during acquisition) was analyzed[102].

## Assessing neuronal activity by in vivo two-photon imaging

*Cranial window implantation and virus injection*. Mice of both sexes were implanted with a cranial window at 3 and 9 months of age (±10 days), respectively and received a stereotaxic injection of the genetically encoded calcium indicator (AAV2/1.hsyn.GCaMP6m.WPRE.SV40 diluted 1:6 in saline—10-month-old cohort or AAV9.syn.jGCaMP7s.WPRE diluted 1:6 in saline—4-month-old cohort (pGP-AAV-syn-jGCaMP7s-WPRE was a gift from Douglas Kim & GENIE Project, Addgene viral prep # 104487-AAV9; RRID:Addgene_104487)[103] into the primary motor cortex (M1)[104]. In brief, mice were first anesthetized with Fentanyl (0.05 mg/kg), Midazolam (5.0 mg/kg), and Metedomidin (0.5 mg/kg). A circular craniotomy with a 2 mm radius, centered at 1.7 mm lateral and 0.8 mm anterior to bregma, was performed, followed by the slow injection of a total of ~1 µl of the calcium indicator into three sites (~300 nl per site at 600 µm cortical depth). A 4 mm round glass coverslip (Warner Instruments) was placed over the cortex and sealed with UV-curable dental acrylic (Venus Diamond Flow, Heraeus Kulzer GmbH). A metal head bar was attached to the skull using dental acrylic (Paladur, Heraeus Kulzer GmbH), allowing for stable positioning during two-photon imaging.

*Two-photon imaging in anesthetized mice*. Four weeks following the cranial window implantation, in vivo two-photon imaging was performed within cortical layer II/III using a two-photon microscope (Hyperscope, Scientifica, equipped with an 8 kHz resonant scanner) at a frame rate of 30 Hz and a resolution of 512 × 512 pixels. Using a ×16 water-immersion objective (Nikon), stacks consisting of 15,000 frames (equivalent to ~8 min) were acquired, covering a field of view (FOV) of 300 × 300 µm. Light source was a Ti:Sapphire laser with a DeepSee pre-chirp unit (Spectra Physics MaiTai eHP)[104]. GCaMP was excited at 910 nm, with a laser power not exceeding 40 mW (typically 10–40 mW). In each mouse, two to five FOVs at cortical depths of 140–310 µm were imaged, yielding 2264 cells in *Fus*[+/+] (n = 25 experiments, 8 mice) and 1107 cells in *Fus*[ΔNLS/+] (n = 13 experiments, 8 mice) at 4 months of age; and 631 cells in *Fus*[+/+] (n = 10 experiments, 5 mice) and 855 cells in *Fus*[ΔNLS/+] mice (n = 14 experiments, 6 mice) at 10 months of age. During imaging, mice were anesthetized with 1.0–1.5 volume % isoflurane in pure O$_2$ at a flow rate of ~0.5 l/min, to maintain a respiratory rate in the range of 110–130 breaths per minute. Body temperature was maintained at 37 degrees using a physiological monitoring system (Harvard Apparatus).

*Image processing and data analysis*. All image analyses were performed in Matlab (Math Works) using custom-written routines[104]. In brief, full frame images were corrected for potential x and y brain displacement, and regions of interests (ROIs) were semi-automatically selected based on the maximum and mean projections of all frames. Fluorescence signals of all pixels within a selected ROI were averaged,

the intensity traces were low pass filtered at 10 Hz. Contamination from neuropil signals was accounted for using the following Eq. (1)[104],

$$\mathbf{F}_{ROI\_comp} = \mathbf{F}_{ROI} - 0.7 \times \mathbf{F}_{neuropil} + 0.7 \times median(\mathbf{F}_{neuropil}) \tag{1}$$

$F_{ROI\_comp}$ stands for neuropil-compensated fluorescence of the ROI, $F_{ROI}$, and $F_{neuropil}$ represent the initial fluorescence signal of the ROI and the signal from the neuropil, respectively. A neuron was defined as 'active' if it displayed at least one prominent calcium transient over 20 frames (corresponding to ~0.7 seconds). The overall difference in the fraction of active cells between 4- and 10-month-old mice could be due to both age as well as the usage of the more sensitive calcium indicator GCaMP7s[105]. To compare the impact of the indicator alone, we also investigated a control 4 m age cohort expressing GCaMP6m, in which case the fraction of active cells was 81% and not different from the average observed in the $Fus^{+/+}$ control cohort used here (ranksum test, $p = 0.87$, 7 experiments in 3 mice).

**Histological techniques.** Male mice were anesthetized with intraperitoneal injection of 100 mg/kg ketamine chlorhydrate (Imalgène 1000®, Merial) and 5 mg/kg xylazine (Rompun 2%®, Bayer), and then transcardially perfused with cold PFA 4% in 0.01 M phosphate buffered saline (PBS). After dissection, brains were post-fixed for 24 h and then included in agar 4% and serial cuts of 40 μm thick were made using vibratome (Leica Biosystems, S2000).

*Peroxidase immunohistochemistry.* For peroxidase immunohistochemistry, sections were incubated 10 min with $H_2O_2$ 3%, rinsed with PBS 1× and incubated with blocking solution (8% Horse serum, 0.3% Bovine Serum Albumin, 0.3% Triton, PBS-0.02% Thimerosal). Sections were rinsed, and then incubated with anti-mouse NeuN or anti-mouse parvalbumin (Millipore, MAB377, 1:100 and Sigma, P3088, 1:1000, respectively) overnight at room temperature. The second day, sections were rinsed and incubated for 2 h at room temperature with biotinylated donkey anti-mouse antibody (Jackson, 715-067-003, 1:500). After sections were rinsed, they were incubated for 1 h in horseradish peroxidase ABC kit (Vectastain ABC kit, PK-6100, Vector Laboratories Inc.), rinsed and incubated with DAB (Sigma, D5905). The enzymatic reaction was stopped by adding PBS 1X, rinsed with water and sections were mounted with DPX mounting medium (Sigma, O6522).

*Quantification.* Images were quantified using a homemade ImageJ plugin. A Region Of Interest (ROI) was first defined by the user as the M1/M2 regions of the cerebral cortex as defined by the Paxinos Atlas[106] using the following coordinates: inter-aural 4.06 mm; Bregma 0.26 mm. For NeuN immunohistochemistry, a second, more anterior region of M1/M2 cortex was also quantified with the following coordinates: Interaural: 5.74 mm, Bregma: 1.94 mm.

In this region, a semi-automated segmentation led to the identification of the labelled structures (cells or nuclei). Finally, the plugin subdivided the previous ROI into 10 subregions and measured either the number of objects per subregion or the proportion of each subregion that is covered by labelled structures.

*Immunofluorescence.* Sections were rinsed with PBS 1X then incubated with blocking solution (8% Goat serum, 0.3% Bovine Serum Albumin, 0.3% Triton, PBS-0.02% Thimérosal) overnight at 4 °C in primary antibody: rabbit anti-FUS antibody (ProteinTech, 11570-1-AP, 1:100) and mouse anti-parvalbumin antibody (Sigma, P3088, 1:1000). After three rinses in PBS, sections were incubated for 2 h at room temperature with Hoechst (Sigma, B2261, 1/50.000) and secondary antibody: Goat anti-mouse Alexa-488 secondary antibody (Invitrogen, A11034, 1:500) and goat anti-mouse Alexa-647 secondary antibody (Invitrogen, A21245, 1:500). Finally sections were subsequently washed with PBS (3 × 10 min) and mounted in Aqua/polymount (Polysciences, 18606).

Images were acquired along the Z axis (Z stacking) using a Zeiss AxioImage.M2 microscope equipped with a Plan-Apochromat ×20/0.8 objective, high performance B/W camera (Orca Flash4, Hamamatsu) and run by the Zeiss Zen2 software. Images were quantified using the ImageJ freeware. First, the user defined ROIs corresponding to the cytoplasm and nucleus or several PV positive cells at several Z positions. Then a homemade macro was used to calculate the ratio, in the green channel, of the cytoplasm intensity divided by the nucleus one.

*Electron microscopy.* Mice from both sexes were used for electron microscopy. Mice were anesthetized by intraperitoneal injection of 100 mg/kg ketamine chlorhydrate and 5 mg/kg xylazine and transcardially perfused with glutaraldehyde (2.5% in 0.1 M cacodylate buffer at pH 7.4). Brains were dissected and immersed in the same fixative overnight. After three rinses in Cacodylate buffer (EMS, 11650), serial cuts of 80 μm thick were made with vibratome. Slides were then post-fixed in 1% osmium in Cacodylate buffer 1 h at room temperature. Finally, tissues were dehydrated in graded ethanol series and embedded in Embed 812 (EMS, 13940). The ultrathin sections (50 nm) were cut with an ultramicrotome (Leica, EM UC7), counterstained with uranyl acetate (1% (w/v) in 50% ethanol) and observed with a Hitachi 7500 transmission electron microscope (Hitachi High Technologies Corporation, Tokyo, Japan) equipped with an AMT Hamamatsu digital camera (Hamamatsu Photonics, Hamamatsu City, Japan). Analysis of electron micrographs was performed as follows: 100 inhibitory synapses located in layers II/III were imaged per animal. Inhibitory synapses were identified as containing at least

one mitochondrion in each synaptic bouton. Synapses morphometry was analyzed using ImageJ freeware (National Institute of Health), where each synaptic boutons' area was manually drawn as previously described[107]. An automated plugin was used to drawn and measure the active zones' length, the number of synaptic vesicles within each bouton and the distance of each vesicle to the active zone, being as the beeline from the vesicle to the active zone. All images were acquired in layer II/III of the M1/M2 regions of the cerebral cortex as defined by the Paxinos Atlas[106] using the following coordinates: interaural 4.06 mm; Bregma 0.26 mm.

**Synaptic density in brain sections.** Male mice were anesthetized by $CO_2$ inhalation before perfusion with PBS containing 4% paraformaldehyde and 4% sucrose. Brains were harvested and post-fixed overnight in the same fixative and then stored at 4 °C in PBS containing 30% sucrose. Sixty micrometers thick coronal sections were cut on a cryostat and processed for free-floating immunofluorescence staining. Brain sections were incubated with the indicated primary antibodies (Rabbit GABAAalpha3 antibody Synaptic Systems, 1:500; Mouse Gephyrin antibody Synaptic Systems, 1:500, Guinea pig VGAT antibody, Synaptic Systems, 1/500) for 48 h at 4 °C followed by secondary antibodies (1:1000) for 24 h at 4 °C. The antibodies were diluted in 1× Tris Buffer Saline solution containing 10% donkey serum, 3% BSA, and 0.25% Triton-X100. Sections were then mounted on slides with Prolong Diamond (Life Technologies) before confocal microscopy.

Confocal images were acquired on a Leica SP8 Falcon microscope using ×63 (NA 1.4) with a zoom power of 3. Images were acquired at a 2048 × 2048 pixel image resolution, yielding a pixel size of 30.05 nm. To quantify the density of synaptic markers, images were acquired in the molecular layer 1/2 of the primary motor cortex area, using the same parameters for all genotypes. Images were acquired from top to bottom with a Z step size of 500 nm. Images were deconvoluted using Huygens Professional software (Scientific Volume Imaging). Images were then analyzed as described[108]. Briefly, stacks were analyzed using the built-in particle analysis function in Fiji[109]. The size of the particles was defined according to previously published studies[77,110]. To assess the number of clusters, images were thresholded (same threshold per marker and experiment), and a binary mask was generated. A low size threshold of 0.01 um diameter and high pass threshold of 1 um diameter were applied. Top and bottom stacks were removed from the analysis to only keep the 40 middle stacks. For the analysis, the number of clusters per 40 z stacks was summed and normalized by the volume imaged (75153.8 $\mu m^3$). The density was normalized to the control group.

**Structural MRI scans.** Male mice were used for MRI studies. All data were acquired on a dedicated small bore animal scanner (Biospec 117/16, Bruker, Ettlingen, Germany) equipped with a cryogenically cooled two-element surface (MRI CryoProbeTM, Bruker BioSpec, Ettlingen, Germany) transmit/receive coil. Anatomical brain images were acquired in coronal slice orientation (30 slices) applying a gradient-echo (FLASH) sequence with acquisition parameters as: TE/TR 2.95/400 ms (TE = echo time, TR = repetition time), matrix 30 × 340 × 340, resolution 250 × 50 × 50 mm³).

Anatomical annotation of brain MRI images was performed in *Fiji*[58] using custom-written routines. In brief, in order to generate a plate corresponding to a single MRI cross section (Figure 4a1) a macro was run to reslice a stack of sagittal plates pursued via the Scalable Brain Atlas[111] according to a manually defined tilting angle by means of the Dynamic Reslice *Fiji* plugin (Figure 4a2). Custom-made plates were then registered onto the corresponding MRI slice by the manual denotation of the major, easily recognizable anatomical landmarks with the Big Warp plugin (Figure 4a3). The thresholding of the warped RGB plates (Figure 4a4) according to the brain structure color code resulted in parcellation of the MRI cross section into single regions. Due to the marked ventriculomegaly, for lateral ventricles and medial septum only, a loss of resolution in the custom plates was noticeable upon warping, therefore these areas along with the entire brain cross section (in order to include olfactory bulbs and cerebellum for the overall intracranial volume) were manually delineated. Finally, region volumes were determined following Cavalieri's principle, i.e., the measurement of the scaled cumulative area was multiplied by the slice increment (Figure 4a5). The volumetric analysis was blinded and evaluated by the same investigator. Code used for volumetric quantification of MRI scans is provided in Supplementary Software.

**RNAseq.** RNAseq on frontal cortex was performed as previously described[10,41]. Briefly, RNA from cortex of 22-months-old male $Fus^{\Delta NLS/+}$ mice and their control littermates were extracted with TRIzol (Invitrogen). RNA quality was measured using the Agilent Bioanalyzer system or RNA screenTape (Agilent technologies) according to the manufacturer's recommendations. Samples were processed using the Illumina TruSeq single Stranded mRNA Sample Preparation Kit according to manufacturer's protocol. Generated cDNA libraries were sequenced using an Illumina HiSeq 2000 sequencer with 4–5 biological replicates sequenced per condition using single read, 50 cycle runs. RNA from the cortex of 5-months-old male $Fus^{\Delta NLS/+}$ and control littermate mice was extracted, and libraries were generated using the Illumina TruSeq single Stranded mRNA Sample Preparation Kit. The cDNA libraries were sequenced on a HiSeq 4000 with three biological replicates per condition using single-end 50 bp read. Total reads sequenced varies from 35 to 45 million reads. Complete QC report will be made publicly available.

Raw reads were mapped to the mouse reference genome GRCm38 with STAR version 2.7.0[112] and default parameters using Ensembl gene annotations (version 87). Gene-level abundance estimates were estimated using the option–quantMode geneCount in STAR. We filtered the lowly expressed genes wherein each gene was required to have at least 15 counts across all samples and used both exonic and intronic reads. The filtered set of genes was used for the PCA plot and differential expression analysis. Differential gene expression analysis was performed with the ARMOR workflow[113] and a cut off FDR value of 0.05 was set in both datasets. RNA samples sequenced in the present study from 5–6-months-old mice ($n = 10$) were pooled to the samples from Sahadevan et al., 2020, ($n = 6$).

**Weighted-gene coexpression network analysis.** Coexpression network analysis was performed using a user-friendly R WGCNA library[114]. We wanted to investigate mouse brain coexpression networks that are disease specific in $Fus^{\Delta NLS/+}$ mice. Biweighted midcorrelations were calculated for all pairs of genes, and then a signed similarity matrix was created. In the signed network, the similarity between genes reflects the sign of the correlation of their expression profiles. The signed similarity matrix was then raised to the power β to emphasize strong correlations and reduce the emphasis of weak correlations on an exponential scale. The resulting adjacency matrix was then transformed into a topological overlap matrix as describe[115]. After scaling the network (consensus scaling quantile = 0.2), a threshold power of 5 was chosen (because it was the smallest threshold that resulted in a scale-free $R^2$ fit of 0.9) and the consensus network was created by calculating the component-wise minimum values for topological overlap. Using 1 − TOM (dissTOM) as the distance measure, genes were hierarchically clustered. Initial module assignments were determined using the blockwiseModules function as follows: blockwiseModules(datExpr, power = 5,TOMType = "signed", minModuleSize = 30, networkType = "signed",deepSplit = 2, reassignThreshold = 0, mergeCutHeight = 0.35, numericLabels = TRUE, pamRespectsDendro = FALSE, saveTOMs = TRUE, verbose = 3). The resulting modules or groups of coexpressed genes were used to calculate the MEs or the first principal component of the module. MEs were correlated with different biological and technical traits like transgenic condition and batch to find disease-specific modules. Module hubs were defined by calculating module membership (kME) values, which are the Pearson correlations between each gene and each ME. Genes with a kME < 0.7 were removed from the module. Network visualization was done with the igraph package in R. Module definitions from the network analysis were used to create synthetic eigengenes for the 1-month, 6 months and 22-months timepoint and were used to understand the trajectory of various modules across timepoints.

**Enrichment analyses using single-cell experiment data.** To reduce false positives, we used FDR-adjusted *P*-values for multiple hypergeometric test comparisons. For cell-type enrichment analysis, we used an already published single-cell mouse brain dataset[116]. Finally, genes in network modules were characterized using EnrichR (version 1.2.5)[117].

**Synaptosomal enrichment followed by RT-qPCR and western blotting.** Frontal cortex was removed from the brains of 4-months-old female mice by micro-dissection, as previously described[118], harvested, rapidly frozen in liquid nitrogen and stored at −80 °C until use. Synaptosomal fraction was isolated using Syn-PER Synaptic Protein Extraction kit (Thermo Scientific, 87793) according to manufacturer's instructions.

On synaptosomal preparations, RNA was extracted using TRIzol reagent (Sigma–Aldrich, 93289). 1 µg of RNA was reverse transcribed using iScript Ready-to-use cDNA supermix (Bio-Rad, 1708841). Quantitative PCR (qPCR) was performed using SsoAdvanced Universal SYBR Green Supermix (Bio-Rad, 172574) and quantified with Bio-Rad CFX Manager software. Gene expression was normalized by computing a normalization factor by Genorm software using three standard genes *Pol2*, *Tbp*, and *Actn* for nervous tissue. Primer sequences are provided in Supplementary Data 2.

For western blotting cytosolic and synaptosomal fractions were prepared using the same protocol, and protein concentration was quantitated using the BCA protein assay kit (Pierce). Fifteen micrograms of proteins were loaded into a gradient 4–20% SDS-PAGE gel (Bio-Rad, 5678094) and transferred on a 0.45 µm nitrocellulose membrane (Bio-Rad) using a semi-dry Transblot Turbo system (Bio-Rad). Membranes were saturated with 10% nonfat milk in PBS and then probed with the following primary antibodies: Anti-Synaptophysin (Abcam, ab14692, 1:1000), Anti-FUS N-ter1 (ProteinTech, 11570, 1:1000), Anti-FUS N-ter2 (Bethyl, A300-293A, 1:2000), and Anti-FUS C-ter (Bethyl, A300-294A, 1:2000) all diluted in 3% nonfat milk in PBS. Blots were washed and incubated with anti-Rabbit secondary antibody conjugated with HRP (P.A.R.I.S, BI2407, 1:5000) for 2 hours. Membranes were washed several times and analyzed with chemiluminescence using ECL Lumina Forte (Millipore, WBLUF0500) using the Chemidoc XRS Imager (Bio-Rad). Total proteins were detected with a stain-free gel capacity and normalized. Uncropped western blot images and stain-free images are provided in supplementary figures.

**Statistics.** If not stated otherwise, data are presented as mean ± standard error of the mean (SEM). Statistical analyses were performed using GraphPad Prism 8

(GraphPad, CA). Unpaired *t*-test was used for comparison between two groups, one-way or two-way analysis of the variance (ANOVA), followed by Tukey's multiple comparison post-hoc test and two-way repeated measures (RM) ANOVA, followed by Sidak multiple comparison post-hoc test were applied for three or more groups. Distributions were compared using the Kolmogorov–Smirnov (KS) test. Results were considered significant when $p < 0.05$.

**Reporting summary.** Further information on research design is available in the Nature Research Reporting Summary linked to this article.

## Data availability
Source data are provided with this paper as supplementary information. The RNAseq datasets that support the findings of this study have been deposited in GEO with the accession codes GSE166615. Source data are provided with this paper.

## Code availability
The code used for MRI analysis is provided as supplementary files.

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

## Acknowledgements

We thank the Imaging Platform of the CRBS (PIC-STRA UMS 38, Inserm, Unistra) and the Plateforme Imagerie In Vitro de Strasbourg for their help in performing imaging for this study. This work was funded by Agence Nationale de la Recherche (ANR-16-CE92-0031 to A.L.B. and L.D., ANR-16-CE16-0015 to L.D., ANR-19-CE17-0016 to L.D.), by Fondation pour la recherche médicale (FRM, DEQ20180339179), Axa Research Funds (rare diseases award 2019, to L.D.), Fondation Thierry Latran (HypmotALS, to L.D. and F.R. and TRiALS to F.R.), Radala Foundation (F.R.), MNDA (Dupuis/Apr16/852-791 to L.D.), ALSA (2235, 3209, and 8075 to L.D. and C.L.T.), Target ALS (to C.L.T.), NINDS/NIH R01-NS108769 (to C.L.T.) Deutsche Forschungsgemeinschaft (DFG, German Research Foundation) under Germany's Excellence Strategy within the framework of the Munich Cluster for Systems Neurology—EXC 2145 SyNergy—ID 390857198 (S.L.), under individual grants no. 431995586, 443642953, 446067541 (F.R.) and under the SonderForschungsBereich (SFB) 1149/2 (251293561 to F.R.), Emmy Noether Programme (S.L.), the Deutsche Gesellschaft für Muskelkranke e.V. (S.L.), and the Graduate School for Systemic Neurosciences GSN-LMU (V.K.). C.L.T. is the recipient of the Araminta Broch-Healey Endowed Chair in ALS. The collaborative work between L.D. and M.P. laboratories was funded by ARSLA (2016). I.S.R. was funded by the Région Grand Est (France).

## Author contributions

J.S.Z. and I.S.R. performed behavioral analysis with help of R.C., L.T., and G.P. I.S.R. and J.S.Z. performed histology, imaging, and electron microscopy with help of P.D.R., M.J., P.K., V.D., and S.D.G. V.K. performed in vivo calcium imaging. S.M. performed RNAseq analysis, with the help of S.S., K.M.H., N.M., C.L.T., and M.P. D.W. performed and analyzed MRI with the help of J.S.Z., H.P.M., S.A., J.K., V.R., A.L., and F.R. S.D. performed synaptosomal extractions, western blotting and RT-qPCR with the help of I.S.R. and G.P. A.L., A.L.B., F.R., M.P., C.L.T., S.L., and L.D. secured funding. M.P., C.L.T., S.L., and L.D. designed and coordinated experiments. J.S.Z., I.S.R., S.L., and L.D. wrote the manuscript.

## Funding

## Competing interests

The authors declare no competing interests.
