## [Peer Review File · Nature Communications]

REVIEWER COMMENTS

Reviewer #1 (Remarks to the Author):

In this manuscript, Scekcic-Zahirovic et al. behaviorally and physiologically characterize a mutant FUS knockin mouse model mirroring the heterozygous genotype of human patients with FUS mutations. Using a suite of behavioral assays, they describe activity, memory, and social phenotypes in mutant mice that develop in an age-dependent manner, which they relate to hyperactivity in the frontal cortex assessed through in vivo calcium imaging. Transcriptomic analysis of frontal cortex from FUS mutant and WT shows altered patterns of gene expression that change over the lifespan of mutant animals and are enriched for genes encoding components of inhibitory synapses.

The findings of this manuscript are novel and of significant interest, since they attest to the pathophysiological effects of disease-associated FUS mutations and their potential phenotypes beyond simply motor neurons disease. The manuscript is well written, and the data are presented clearly. In addition, there are several exciting findings emphasizing the potential role of FUS in maintaining neuronal networks that control behavior. Even so, as written, the manuscript is a collection of observations without clear attempt to discern a mechanism for FUS in the development of observed phenotypes (behavioral changes, differentially expressed genes, neuronal activity, ventriculomegaly), and no clear connection between these. These and other concerns are listed below:

Major comments:

- The changes visualized by MRI, including brain atrophy but no change in cortical thickness, are intriguing. If not due to cortical thinning, how does ventricular enlargement occur? Hydrocephalus is not uncommon in laboratory mouse strains, but another possibility is white matter thinning. This is an interesting and impressive phenotype that has no explanation in the current work.
- The demonstration of cortical hyperactivity is also very interesting, but as it stands this is limited to 10 mo old animals, despite the fact that behavioral abnormalities were detected as early as 4 mos of age. How do these observations fit with the behavioral changes or differentially expressed genes?
- The RNA-seq analysis showed substantial differences in gene expression at 5 months of age in cortex (Fig. 5b), in distinct disagreement with the co-submitted manuscript that showed very few differences in forebrain (Supp. Fig. 4g). The reason for this discrepancy is unclear.
- The changes in FUS localization in PV neurons are intriguing, but without examining other cell types it is unclear if these age-dependent differences are unique to PV neurons or not
- Similarly, the ultrastructural changes in inhibitory synapses could be very important, but it is unclear whether other synapse types may display similar or even more profound changes in FUS Δ NLS mice.
- The impact of increased RNA levels (for some inhibitory genes) within synaptosomal preparations is unclear. What consequences does this have for translation and other downstream events, particularly since many genes that show expression changes were not affected? How might FUS be involved here?
- How do the authors' data on hyperexcitability mediated by interneuron dysfunction in FUS Δ NLS mice relate to previous data demonstrating hypoexcitability in human iPSC-derived neurons carrying FUS mutations (PMID: 26946488)?
- In these analyses, it is difficult to tell primary from secondary events in pathogenesis. For instance, the reduction in expression of inhibitory synapse RNAs could be primary (i.e. due to sequestration by mutant FUS) or secondary (i.e. due to PV neuron dysfunction or reductions in inhibitory synapses). The authors provide little data to help distinguish among these possibilities, but data from the

accompanying manuscript suggest that direct binding by mutant FUS to essential transcripts involved in inhibitory synapse maintenance is unlikely to be responsible.

- As the behavioral, physiological imaging, RNA-seq, and staining experiments are done at different timepoints, a timeline of when each study was done would greatly benefit the reader.

Minor comments:

- Why do some behavioral deficits (Fig. 2f) disappear by 22 months?
- Line 149: the title to this section is 'executive dysfunction in FUS Δ NLS mice' but the authors test spatial memory and extinction, rather than executive functioning. The same is also true for line 327, in the discussion.
- Line 181: these results do not 'confirm social disinhibition' but instead are consistent with this idea.
- Line 194: "unraveled" seems awkward here. Perhaps "uncovered"?
- Line 197: in the section on "Increased spontaneous neuronal activity in Fus Δ NLS/+ mice in vivo", to help readers understand the experiment the authors should describe the generation of FUS Δ NLS/GCaMP6m mice
- Also, it is unclear whether hyperactivity was limited to this brain region (layers II/III of the frontal cortex)
- The plots in Fig. 3d-f make it difficult to assess distribution. Instead, the authors are encouraged to show violin plots, density plots, or histograms.
- In Fig. 7a, it would help to have a WB of total lysate to be able to compare FUS Δ NLS levels in specific compartments (cytoplasm, synaptosome) to total FUS levels, since the loss of one normal FUS allele might be expected to produce an overall reduction of full-length FUS
- Do the observations regarding neurexins in PV neurons suggest that replacing these genes specifically in PV neurons might be sufficient to rescue phenotypes in the FUS Δ NLS mice?

Reviewer #2 (Remarks to the Author):

The manuscript by Scekcic-Zahirovic, Sanjuan-Ruiz et al describes a role for the RNA binding protein FUS in the synapse. They describe the behavioural characterization of their previously published Fus Knock In (KI) mouse model carrying the equivalent of an ALS causative mutation that affects its nuclear localization signal (NLS), together with data showing that mutant FUS increase its localization at synaptic sites, and that the mutation produces cortical hyperexcitability in vivo, all underlined by changes in the synaptic transcriptome.

Overall, the data presented is generally of good quality and would be suitable for publication after addressing the following points:

1. According to the materials, the behavioural characterization of the mice was carried out only in males. This is not standard procedure and is not in accordance with ARRIVE guidelines. Is there a particular reason for this? Actually, in some analysis it is clear that both sexes have been used (for example in the in vivo recordings), whereas in other work it is not entirely clear (for example in RNA-seq, immunohistochemistry or biochemical analysis). This should be made more obvious. Authors should also explain the rationale for using females in some of the analysis after not been used for behaviour.
2. It is not clear how the mice were maintained, nor which C57BL/6 substrain was used. I assume the mouse cohorts were maintained by backcrossing heterozygous mutant carriers to C57BL/6 mice? It should be clear which C57BL/6 substrain was used, C57BL/6J? Also, the correct nomenclature is

C57BL/6 (plus substrain) not C57/BL6 as in the manuscript.

3. It is also not clear how the mice were maintained for the behavioural analysis. Male mice were used in some tests that required them being housed individually, but were they housed back together afterwards? or kept individually? This should be made more obvious as it might actually affect the results.

4. In terms of the behavioural analysis, for the resident intruder analysis, the authors only present data for the social interactions, but what about the aggressive behaviours? Were there any differences between genotypes?

5. According to the materials, RNA-seq appear to have been done using different platforms for the early (5 months) and late (22 months) time points. If this is the case, this should be made clearer and also should discuss the possible implications particularly for the comparisons between the different timepoints.

6. About the immunohistochemistry analysis at synapses, how specific are the markers used for PV interneurons? This should be made clear. Also, were any of the markers used changed at the RNA-seq level?

7. About the increased levels of FUS protein at synaptosomal fractions, authors should include controls quantifying FUS levels from total lysates, including nuclear and cytoplasmic fractions. Is the increase of FUS at synapses simply reflecting higher levels of FUS overall in mutant mice due to auto-regulation? Or is there a specific increase of FUS at the synapses?

8. Finally, in the section "Defects in inhibitory neurons in *Fus Δ NLS/+* mice" on line number 256 it should refer to Figure 5C, not Figure 5D.

Reviewer #3 (Remarks to the Author):

FUS-ALS, caused by C-terminal FUS mutations leading to FUS cytoplasmic mislocalisation, constitutes the most severe form of ALS, and although rare, is representative of RNA binding protein dysfunction/misregulation very commonly observed in ALS and FTD. Furthermore, FUS mislocalisation has also been observed in non-FUS-mutated cases of ALS, while FUS cytoplasmic mislocalisation is observed in a subset of FTD cases, albeit without mutations in most cases. Taken together, the findings presented in this paper are of relevance to the wider field of ALS/FTD. FUS is known to be a dosage sensitive gene, and while other, transgenic, animal models of FUS-ALS display stronger phenotypes, the knock-in model used in this study provides better physiological relevance due to the endogenous expression levels (and endogenous control of expression) of mutated FUS.

The authors have previously investigated the spinal cord and motor function in this model, and here investigate behavioral phenotypes and brain dysfunction, given the links of FUS to FTD. As a ubiquitously expressed gene, it is important to understand the wider effects (beyond motor neurons) of FUS mislocalisation to fully understand FUS-related disease – and in this respect the authors provide strong overall evidence of an inhibitory cortical neuron synaptopathy.

Behavioral analyses show relevant phenotypes (although I have some reservations and queries) that nicely tie in with cortical synaptic dysfunction - specifically inhibitory synapse function - to provide a mechanistic explanation. Considering the mouse utilized is a physiological model, and that mice have a relatively short lifespan, a lack of neurodegeneration is not too surprising and should not be considered problematic. The mechanisms described provide clues regarding early pathogenic changes critical to understanding and treating FUS-related disease.

Behavioral analyses

The authors ran a series of behavioural tests on male mice. Were female mice not at all tested, or did they not show a phenotype?

Spontaneous locomotor activity was increased, similar to findings from previously reported FUS models. There seems to be some confusion with the statements: "Hyperactivity was not caused by reduced anxiety," and later "hyperactivity did not appear to be associated with increased anxiety". Either way (I think reduced anxiety is more likely to confound hyperactivity), the authors try to rule out anxiety as a confounding factor, but only test this at 10 months, when hyperactivity is not present (no reduced central time in open field, little evidence of daytime hyperactivity in home cage analysis). I therefore feel the authors should not explicitly rule out the possibility of an anxiety phenotype, in particular at the early 4 month timepoint when hyperactivity is most prevalent. If hyperactivity diminishes as the mice age – could the same age-related process not have the same effect on an anxiety effect?

Behavioural changes in FTD are complex and include a range of symptoms such as hyperactivity and anxiety, but also apathy and disinhibition. As such I'm not sure why it would be necessary to rule out confounding factors that may in fact be relevant to disease, and may yield from a common underlying synaptic dysfunction. In short, I would not claim to rule out an anxiety phenotype linked to hyperactivity, simply that you saw no evidence for it in aged mice.

Executive dysfunction was assessed using MWM, where I feel claims regarding "poor performance" are overplayed. Mutants find the platform in the same timeframe as WTs, and while they do spend a statistically significant greater amount of time in random quadrants versus the target quadrant, the actual difference in time is marginal. Similarly, the statement "Fus Δ NLS/+ mice extinguished their previous memory much faster than wild type mice" is an overstatement (Fig 1g). There is an effect, but it seems penetrance and the size of effect is very low.

Sociability was measured in 2 distinct tests at 3 time points. The resident intruder paradigm consistently showed - at 4, 10 and 22 months - mutants spending more time interacting with the intruder. The 3-chamber test only showed increased sociability at 10 months, but not at the earlier timepoints. Can the authors speculate on why there may be a discrepancy between tests? Do the authors think increased sociability could be linked to hyperactivity and impulsivity?

Neuronal activity

The authors clearly demonstrate increased neuronal activity in vivo, in the frontal cortex, in 10 month old mice.

MRI

MRI imaging showed ventricular enlargement in 12 month old mutant mice, but was not due to changes in the cortex or neuron loss. Authors state that this could be due to "neuronal dysfunction," but is there a precedent for this? Could the morphological changes be developmental in origin, since cases of FUS-ALS are often juvenile in nature and developmental abnormalities have been reported in the literature (e.g. developmental delay)? Indeed, at least 2 developmental genes - Dlx5 and Dlx6 - were picked up in transcriptomics.

Transcriptomics

Frontal cortex RNAseq was performed at 5 and 22 months. Authors describe an intriguing predominance of upregulation of genes at 5 months and downregulation at 22 months – and GABA related genes in particular were first upregulated then downregulated, pointing to inhibitory neuron dysfunction, tying in with neuronal hyperexcitability. Comparison was made to previous datasets, including FUS^{-/-} mice. It has been shown that transcriptional changes associated with FUS mutation are consistent with loss of function (i.e. changes strongly overlap in the same direction with FUS^{-/-} mice/cells). How do the results here chime with previous datasets in terms of directionality of gene changes – i.e. do changes in gene expression here chime with loss of function at early or late timepoints, or do the authors find novel gain of function changes?

Defects in inhibitory neurons

Inhibitory cortical PV interneurons were further investigated in light of the observed transcriptomic changes, and markers associated with this population were found to be altered, and inhibitory synapse markers were also shown to be altered at the protein level.

Of note, ultrastructural analyses showed dysmorphology of inhibitory cortical synapses, together with the observation that inhibitory synaptic density was reduced. While these were only shown at 22 months, in the accompanying paper inhibitory synapse defects were shown at early timepoints; altogether, suggestive of causation with respect to neuronal hyperexcitability.

Abnormal synaptosomal fractions

Finally, synaptosomal fractions were analysed via qPCR and Western Blotting in 5 month old mice. Evidence for mutant, but not WT, FUS protein was found to be enriched at synapses – as might be expected due to lack of nuclear localization signal on the mutant protein. At the mRNA level, 3 FUS binding mRNAs: *Fus*, *Nrxn1*, and *Gabta1*, were also enriched at the synapse in mutants (the latter 2 involved in inhibitory synapse function). Can the authors please discuss further, here or in the discussion, the potential mechanism of *Nrxn1* and *Gabta1* mRNA increases (i.e. how could mutant FUS cause this?), and how increases in the mRNAs fit in with their hypothesis of reduced inhibitory synapse function?

Can the authors please explain why they chose to look at *Chrna7*, and what the significance might be?

Discussion

In the sentence, "...we observed an early and sustained spontaneous locomotor hyperactivity in *Fus*^{ΔNLS/+} mice" - I don't agree with "sustained", as at 10 months, hyperactivity levels were far more restricted, and not at all present from the open field test; and the later 22 month time point was not performed.

As stated above, I think the authors cannot rule out anxiety being linked to hyperactivity.

Final comments

In summary, I find the work as a whole to convincingly show that physiological levels of FUS mislocalisation can lead to inhibitory synapse defects and behavioural phenotypes linked to increased cortical activity that may be of important relevance to the wider field of ALS/FTD, and will likely spur further investigation in other mouse models, and patients.

Thomas Cunningham
MRC Harwell, Oxfordshire, UK

Scekic-Zahirovic, Sanjuan-Ruiz and collaborators
Detailed response to reviewers

Reviewers comments in black
Our responses in blue.

Reviewer #1 (Remarks to the Author):

In this manuscript, Scekic-Zahirovic et al. behaviorally and physiologically characterize a mutant FUS knockin mouse model mirroring the heterozygous genotype of human patients with FUS mutations. Using a suite of behavioral assays, they describe activity, memory, and social phenotypes in mutant mice that develop in an age-dependent manner, which they relate to hyperactivity in the frontal cortex assessed through in vivo calcium imaging. Transcriptomic analysis of frontal cortex from FUS mutant and WT shows altered patterns of gene expression that change over the lifespan of mutant animals and are enriched for genes encoding components of inhibitory synapses.

The findings of this manuscript are novel and of significant interest, since they attest to the pathophysiological effects of disease-associated FUS mutations and their potential phenotypes beyond simply motor neurons disease. The manuscript is well written, and the data are presented clearly. In addition, there are several exciting findings emphasizing the potential role of FUS in maintaining neuronal networks that control behavior. Even so, as written, the manuscript is a collection of observations without clear attempt to discern a mechanism for FUS in the development of observed phenotypes (behavioral changes, differentially expressed genes, neuronal activity, ventriculomegaly), and no clear connection between these. These and other concerns are listed below:

Major comments:

- The changes visualized by MRI, including brain atrophy but no change in cortical thickness, are intriguing. If not due to cortical thinning, how does ventricular enlargement occur? Hydrocephalus is not uncommon in laboratory mouse strains, but another possibility is white matter thinning. This is an interesting and impressive phenotype that has no explanation in the current work.

We thank the reviewer for this comment.

We first sought to determine whether the lack of cortical atrophy could be due to imprecise volume measurement. To this end, we set up a new quantification method of mouse MRI data (code included as supplementary material), that allowed us to semi-automatically extract volumes of many more brain regions than initially considered. We first provide results showing normal intracranial volume (ICV; it represents the overall volume occupied by the brain, including the cerebellum and the brainstem, and any internal structure such as the ventricles) in *Fus^{ΔNLS/+}* mice. Since ICV can slightly change from mouse to mouse (due to normal biological variations), we normalized the volumes of cerebral substructures to the ICV of each mouse. When we separated the ventricular volume and the parenchymal fraction (comprehensively including all grey and white matter, besides the ventricles) we could confirm a significant increase (almost doubling) of the ventricular volume, corresponding to the comparable decrease in parenchymal fraction. This rules out acute hydrocephalus and points toward “ex-vacuo” hydrocephalus. We confirmed the unchanged volume of neocortex, a result pointing toward the subcortical origin of the hydrocephalus. In fact, we provide evidence of decreased volumes in subcortical structures including medial septum, claustrum and amygdalar nuclei. We do not have evidence of atrophy in hippocampus or dorsal striatum, suggesting the involvement of mainly basal forebrain structures.

Of note, the T1-weighted MRI sequence used is relatively insensitive to white matter, and thus is not appropriate to precisely quantify a potential loss of white matter, which thus cannot be excluded.

We have included these new results, the code for the volume quantification, and revised the discussion accordingly.

- The demonstration of cortical hyperactivity is also very interesting, but as it stands this is limited to 10 mo old animals, despite the fact that behavioral abnormalities were detected as early as 4 mos of age. How do these observations fit with the behavioral changes or differentially expressed genes?

To answer the reviewer's comment, we performed additional experiments and bioinformatic analysis.

We first performed in vivo two-photon imaging of cortical neuronal activity at an earlier time point of 4 months of age. Interestingly, and very much in line with the behavioral results, we observed an intermediate phenotype: 4 months old *Fus^{ΔNLS/+}* mice displayed increases in transient frequency, similar to 10 months old mice, but the fraction of active neurons did not differ at that age.

Intriguingly, we observed a significant overall difference in the fraction of active cells between 4 and 10 month old mice irrespective of genotype. This could be due to both age as well as the usage of the more sensitive calcium indicator GCaMP7s¹ in these new experiments. To compare the impact of the indicator alone, we also investigated a control 4m age cohort expressing GCaMP6m, in which case the fraction of active cells was 81% and not different from the average observed in the *Fus^{+/+}* control cohort used here (ranksum test, $p = 0.87$, 7 experiments in 3 mice). These data have been included in a revised Figure 3 and demonstrate a relevant correlation between electrophysiological alterations and altered behavior.

To further understand the relationship between behavior, electrophysiological alterations and gene expression, we used a systems biology approach to analyze simultaneously the transcriptomes obtained in this study (5 and 22 months of age), and those of the companion manuscript (1 and 6 months of age). As can be seen from the revised Figure 5 (see below for details of the bioinformatic approach), we observed a progressive alteration of gene expression between 1 and 5-6 months that is also consistent with altered behavior.

Overall, we found that mainly inhibitory synapses are affected in the mouse model, which fits very well with the hyperexcitability and increased neuronal activity levels we observe in vivo and aligns with the observed behavioral deficits.

Summarizing, our new results and analysis demonstrate that the altered behaviors are well correlated with the progressive alterations in electrophysiological alterations and gene expression in mutant mice.

- The RNA-seq analysis showed substantial differences in gene expression at 5 months of age in cortex (Fig. 5b), in distinct disagreement with the co-submitted manuscript that showed very few differences in forebrain (Supp. Fig. 4g). The reason for this discrepancy is unclear.

We would like to thank the reviewer for this insightful comment highlighting this discrepancy that we did not notice beforehand. We agree that it is critical to understand the causes of such differences.

In order to identify the origin of this discrepancy, we exchanged datasets and pipelines of analysis between both labs. This revealed that the different outputs between both studies did not lie in biological differences, but purely on different bioinformatic procedures applied in the investigation. We previously used a DEseq based model, that is classically used, while

Sahadevan and collaborators rather used a more recent and more conservative EdgeR based model.

This difference between both pipelines is illustrated in the following QQ plots, that show the correlation between expected and observed p-values for the same dataset using either our pipeline (left) or Sahadevan et al pipeline (right). The ideal calibration is expected to lie within the grey zone.

As a matter of fact, our pipeline (left) tended to inflate the observed p-value as compared to expected p-values. This likely resulted in an overestimation of the number of differentially expressed genes as compared to the Sahadevan pipeline which appeared to be better calibrated on p-values although mildly underpowered. As a consequence, our bioinformatic analysis tended to identify more differentially expressed genes, and was thus more sensitive to false positives (type I error), while contrastingly, Sahadevan et al pipeline was more stringent, and could lead to a number of differentially expressed genes that would not be identified as significant (type II error). Our less conservative approach allowed us to pinpoint globally affected pathways (synaptic physiology, mostly inhibitory), and enabled us to reason in a more “systems biology” approach.

In order to avoid the discrepancy between both papers, we decided to use the exact same conservative pipeline of analysis for the identification of differentially expressed genes between both manuscripts (used previously in the Sahadevan manuscript).

Using Sahadevan pipeline, we replicated their findings using our own dataset, and did not identify differentially expressed genes. We then used a systems biology approach, called WGCNA (weighted gene coexpression network analysis), classically used to study co-expressed genes, clustered in modules, and define network nodes, defined as hub genes. Importantly, the WGCNA approach does not require to *a priori* identify differentially expressed genes and uses the whole dataset for analysis.

Using WGCNA, we identified two mRNA modules (labelled as Turquoise and Yellow modules) significantly correlated with the genotype condition in the cortex. The turquoise module is globally downregulated, while the yellow module is upregulated (yellow) in *Fus^{ANLS/+}* mice (revised Figure 5a). Importantly, the turquoise module is enriched in neuronally expressed genes (revised Figure 5b), and gene ontology analyses demonstrates it is a module gathering genes involved in synaptic physiology (Figure 5d,e,f). Importantly, many of the hub genes identified in the turquoise module are directly or indirectly related to GABAergic

neurotransmission, and includes GABA receptors (*Gabrb1*), and key molecules involved in development of the GABAergic synapse (*Nrxn1*, *Dmd*), thus confirming our previous analysis of these data. However, this new approach also suggests a more widespread defect of synapses, since GO Terms related to glutamatergic or cholinergic synapse are also enriched in the Turquoise module. Thus, we adapted the result section and re-organized our manuscript to take into account this new information.

Summarizing, we would like to thank the reviewer for pointing out this discrepancy between our jointly submitted manuscripts. The harmonization of the analytical bioinformatic pipelines allowed us to reach the same results between both laboratories, and the use of systems biology approaches confirmed, independently, our previous interpretation of these transcriptome results.

All these changes have been implemented in the revised manuscript in the corresponding results section and figures.

- The changes in FUS localization in PV neurons are intriguing, but without examining other cell types it is unclear if these age-dependent differences are unique to PV neurons or not

From our previous studies, we can state that the differential mislocalization of FUS in PV neurons is not unique to this cell type. We found altered FUS mislocalization in (i) motor neurons, (ii) oligodendrocytes, and (iii) skeletal muscle in these same mice²⁻⁴. Mislocalization of FUS is actually a direct consequence of the genetic manipulation performed.

To take into account the reviewer's comment, we modified the text of this results section highlighting that such mislocalization was not unique to inhibitory neurons: "As a result of the Δ NLS mutation, FUS would be expected to accumulate in the cytoplasm of PV neurons as previously shown in other cell types²⁻⁴. We thus performed double immunostaining for FUS and parvalbumin, and determined the nuclear/cytoplasmic ratio selectively in PV neurons."

- Similarly, the ultrastructural changes in inhibitory synapses could be very important, but it is unclear whether other synapse types may display similar or even more profound changes in FUS Δ NLS mice.

We thank the reviewer for this comment. We would like to point out that it was not our intention to claim for an exclusive, cell specific, defect in inhibitory synapses. Part of the discussion section was previously focused on discussing whether there could also be defects in excitatory neurons.

To answer reviewer's comment, we performed ultrastructural analysis of excitatory synapses in the same animals in which analysis of inhibitory synapses was performed. We observed that excitatory synapses were also affected, albeit to a lesser extent than inhibitory synapses. Interestingly, abnormalities in synaptic ultrastructure were opposite in excitatory compared to those previously observed in inhibitory synapses: bouton size, synaptic vesicle number and synaptic active zone length appeared slightly smaller in mutant mice compared to wild type animals. These data now show that synaptic defects, although predominantly affecting inhibitory synapses, are widespread in the cortex of these animals, and are entirely consistent with our new WNCGA analysis of transcriptome, performed in response to a previous comment. The manuscript has been revised accordingly to include these new results in revised figure 6 and accompanying text.

- The impact of increased RNA levels (for some inhibitory genes) within synaptosomal

preparations is unclear. What consequences does this have for translation and other downstream events, particularly since many genes that show expression changes were not affected? How might FUS be involved here?

We agree with the reviewer that this is a major question. We however believe that resolving this question is beyond the scope of the current manuscript and should be the focus of follow up studies.

The companion manuscript by Sahadevan and collaborators provide in this case at least one possible mechanism of FUS involvement: they show that the Δ NLS mutation increases RNA stability of at least a large fraction of FUS synaptic targets. It is thus likely that the increased cytoplasmic FUS content alters synaptic homeostasis through modified synaptic RNA stability either through direct RNA binding or indirectly.

To take into account the reviewer's comment, we modified our discussion in order to highlight this hypothesis based on the results of the companion manuscript and clearly state this limitation of our current study.

- How do the authors' data on hyperexcitability mediated by interneuron dysfunction in FUS Δ NLS mice relate to previous data demonstrating hypoexcitability in human iPSC-derived neurons carrying FUS mutations (PMID: 26946488)?

We thank the reviewer to point out to us this interesting study. We do not have a clear and straightforward answer to relate these two studies. It is noteworthy that we found hyperexcitability in pyramidal neurons in an in vivo model in a highly integrated context, while this study focuses on a different cell type (motor neurons) studied in culture conditions. Interestingly, there is ample evidence that upper motor neurons in cortex behave differently than lower motor neurons in spinal cord. While UMNs turn hyperexcitable (at least during early stages of the disease), lower motor neurons have been shown to become hypoexcitable in vivo prior to denervation⁵. Thus, in addition to the very different methodological approaches used in the mentioned study and ours, these studies strongly differ with respect to the neuronal population studied. For a comprehensive review on that topic please see⁶.

- In these analyses, it is difficult to tell primary from secondary events in pathogenesis. For instance, the reduction in expression of inhibitory synapse RNAs could be primary (i.e. due to sequestration by mutant FUS) or secondary (i.e. due to PV neuron dysfunction or reductions in inhibitory synapses). The authors provide little data to help distinguish among these possibilities, but data from the accompanying manuscript suggest that direct binding by mutant FUS to essential transcripts involved in inhibitory synapse maintenance is unlikely to be responsible.

We agree with the reviewer that our current study does not fully allow to disentangle primary from secondary events, especially regarding cell type specific effects of the mutation.

In general, demonstrating whether rescuing the mutation in specific neuronal subsets (in particular in PV neurons) requires time consuming cross breeding, and longitudinal follow up of large cohorts of double transgenic mice. Such experiments, currently ongoing in our laboratories, require >2 years to be completed, and are in our opinion beyond the scope of the current manuscript.

To address the reviewers comment, we rephrased and developed our discussion in alignment with the Sahadevan companion manuscript, which demonstrates that the FUS protein binds to a number of synaptic mRNAs is involved in regulating the stability of at least a subset of them, thus possibly underlying the observed effects.

- As the behavioral, physiological imaging, RNA-seq, and staining experiments are done at different timepoints, a timeline of when each study was done would greatly benefit the reader.

In the revised manuscript, we now include a supplementary figure summarizing the different experiments and helping the reader to better understand our studies.

Minor comments:

- Why do some behavioral deficits (Fig. 2f) disappear by 22 months?

The performance of wild type mice decreases with age in many of the tests, in which case the difference to the FUS transgenic line is abolished. This is a likely explanation for the lack of some behavioral defects in aged mice.

- Line 149: the title to this section is 'executive dysfunction in FUS Δ NLS mice' but the authors test spatial memory and extinction, rather than executive functioning. The same is also true for line 327, in the discussion.

We corrected this in the revised manuscript.

- Line 181: these results do not 'confirm social disinhibition' but instead are consistent with this idea.

We corrected this in the revised manuscript.

- Line 194: "unraveled" seems awkward here. Perhaps "uncovered"?

We corrected this in the revised manuscript.

- Line 197: in the section on "Increased spontaneous neuronal activity in Fus Δ NLS/+ mice in vivo", to help readers understand the experiment the authors should describe the generation of FUS Δ NLS/GCaMP6m mice

We edited the text of the revised manuscript and added information regarding AVV-mediated expression of the genetically encoded indicator .

- Also, it is unclear whether hyperactivity was limited to this brain region (layers II/III of the frontal cortex)

We only investigated this region and cannot exclude that hyperactivity could also be observed in other brain regions. We, however, expanded our investigations to now also include a younger 4month old cohort. Future experiments are needed to also probe neuronal dysfunction in other brain areas.

- The plots in Fig. 3d-f make it difficult to assess distribution. Instead, the authors are encouraged to show violin plots, density plots, or histograms.

We corrected this in the revised manuscript.

- In Fig. 7a, it would help to have a WB of total lysate to be able to compare FUS Δ NLS levels in specific compartments (cytoplasm, synaptosome) to total FUS levels, since the loss of one normal FUS allele might be expected to produce an overall reduction of full-length FUS

This control is now included in the revised manuscript.

We performed western blotting on the total fractions of these experiments, and observed, as expected, an increase in total FUS levels, resulting from autoregulation. Interestingly, however, the relative increase in FUS in total extracts was about 3 times, while we observed almost 10 times increased FUS levels in synaptosomal fractions, thus arguing for a selective synaptic accumulation of FUS in these mutant mice. Of note, this synaptic accumulation of mutant FUS is also observed in independent experiments, using different protocols, in the companion manuscript of Sahadevan and collaborators.

- Do the observations regarding neurexins in PV neurons suggest that replacing these genes specifically in PV neurons might be sufficient to rescue phenotypes in the FUS Δ NLS mice?

This is an intriguing, but difficult to address, question, especially because of the very complicated biology of neurexins and neuroligins. We now introduced a sentence in the revised manuscript, suggesting that this work could be the basis to identify potential mechanisms to be targeted in future preclinical research (including neurexins): , and whether rescuing synaptic defects in inhibitory neurons might translate into an efficient therapeutic strategy.

Reviewer #2 (Remarks to the Author):

The manuscript by Scekcic-Zahirovic, Sanjuan-Ruiz et al describes a role for the RNA binding protein FUS in the synapsis. They describe the behavioural characterization of their previously published Fus Knock In (KI) mouse model carrying the equivalent of an ALS causative mutation that affects its nuclear localization signal (NLS), together with data showing that mutant FUS increase its localization at synaptic sites, and that the mutation produces cortical hyperexcitability in vivo, all underlined by changes in the synaptic transcriptome. Overall, the data presented is generally of good quality and would be suitable for publication after addressing the following points:

1. According to the materials, the behavioural characterization of the mice was carried out only in males. This is not standard procedure and is not in accordance with ARRIVE guidelines. Is there a particular reason for this? Actually, in some analysis it is clear that both sexes have been used (for example in the in vivo recordings), whereas in other work it is not entirely clear (for example in RNA-seq, immunohistochemistry or biochemical analysis). This should be made more obvious. Authors should also explain the rationale for using females in some of the analysis after not been used for behaviour.

We thank the reviewer for this important question and now provide details on genders used in all experiments in both materials and methods and figure legends.

Most of the experiments, as correctly noted by the reviewer, have been performed in male mice, in particular all behavioral analysis. Female mice were only included for (i) electron microscopy (in a gender balanced group), (ii) in vivo recordings (in a gender balanced group), and (iii) for synaptosomal preparations.

We fully agree that ideally both sexes should be included for behavioural experiments. However, these experiments were carried out as part of a larger ethical application that included also the study of homozygous Δ NLS mice². In this context, most of the female *Fus* ^{Δ NLS/+} mice were used for breeding purposes. In order to minimize the numbers of animals used in these experiments, and minimize the numbers of animals generated in the overall project, and thus adhere as strictly as possible to the 3R rules, our animal license only included behavioural analysis in male mice (Cremeas ethical committee reference number AL/27/34/02/13).

In addition, a large number of studies have documented gender differences differences in the prevalence of FTD. With respect to FTD, Ratnavalli and colleagues⁷ found that men were four times more likely than women to be affected by bvFTD. Johnson and colleagues also reported sex differences in clinical presentation of FTD⁸. While it is currently not known whether there could be substantial differences in FTD-FUS, we think that the investigation of possible gender differences in the phenotypes of *Fus* ^{Δ NLS/+} mice goes beyond the current manuscript and would require the generation of parallel cohorts of male and female mice. However, in several cross breeding experiments that have been performed in other contexts than the current study, we observed similar defects in female *Fus* ^{Δ NLS/+} mice, at least in motor behaviors (inverted grid), and we did not observe major differences between male and female mice in the experimental read-outs in which both sexes have been studied (in vivo recordings, electron microscopy).

To take into account the reviewer's comment:

- we have reviewed our manuscript in order to make more explicit in each figure the sex of the mice being investigated.
- It is now explicitly stated in the materials and methods section that we could not study both male and female mice for practical and ethical reasons to minimize the number of mice generated for these studies.

- The discussion section was also modified to explicitly state that possible sex differences might exist and have not been studied here.

2. It is not clear how the mice were maintained, nor which C57BL/6 substrain was used. I assume the mouse cohorts were maintained by backcrossing heterozygous mutant carriers to C57BL/6 mice? It should be clear which C57BL/6 substrain was used, C57BL/6J? Also, the correct nomenclature is C57BL/6 (plus substrain) not C57/BL6 as in the manuscript.

We apologize for the error: the substrain used was C57Bl/6J. This is now corrected in the manuscript.

3. It is also not clear how the mice were maintained for the behavioural analysis. Male mice were used in some tests that required them being housed individually, but were they housed back together afterwards? or kept individually? This should be made more obvious as it might actually affect the results.

We apologize for omitting to formulate this more precisely: Until the mice reached the age when the behavioral tests were performed mice were group-housed. Once mice were single-housed for the behavioral task they were kept individually for only a period necessary to finalize the set of behavioral experiments and in order to minimize possible negative effects of isolation, afterwards cohorts were sacrificed and processed for downstream analyses. This is now corrected in the manuscript.

Since actimetry by default requires to separate mice into individual cages for 3 consecutive days and resident intruder and three chamber tests are preceded by 1 week of isolation, adult mice could not be re-grouped without risk of increased aggressive behavior.

4. In terms of the behavioural analysis, for the resident intruder analysis, the authors only present data for the social interactions, but what about the aggressive behaviours? Were there any differences between genotypes?

We thank the reviewer for this comment. We reanalyzed our videos differentiating aggressive behaviours (attacks) and social interactions.

We only observed attacks at 4 months of age. No attacks were observed in 10 or 22 months old groups.

There was no difference in the total attack time between genotypes (13.0s +/-1.4 s in +/+ mice vs 11.6 +/- 1.0 s in Δ NLS/+ mice, $p = 0.88$ Unpaired Student's t-test). There was also no difference in the latency before the first attack or the duration of attacks. Thus, in this test, and at that age, we did not observe significant difference in aggressive behaviours in Δ NLS/+ mice. These data were included in revised version of our manuscript.

5. According to the materials, RNA-seq appear to have been done using different platforms for the early (5 months) and late (22 months) time points. If this is the case, this should be made clearer and also should discuss the possible implications particularly for the comparisons between the different timepoints.

The reviewer is right in his/her comment: the previous version of the manuscript included RNAseq performed on two different platforms. The revised version of the manuscript further integrates these previous datasets with those of Sahadevan and collaborators (co-submitted manuscript) in order to allow the use of WGCNA, requiring a relatively large number of samples. To render this comparison meaningful despite the different platforms, we removed potential batch effects using a negative binomial regression model to estimate batch effects

based on the count matrix, according to a published method⁹. This is illustrated in Figure S3c of the revised manuscript. Efficiency of the method is shown in Figure S3d, demonstrating the clustering between genotypes after removal of the batch effects.

This is now stated in the results section, page 9, as well as in methods section page 24.

6. About the immunohistochemistry analysis at synapses, how specific are the markers used for PV interneurons? This should be made clear. Also, were any of the markers used changed at the RNA-seq level?

The markers used (VGAT, Gephyrin, GABA_AR α 3) are not specific to PV interneurons, but found at all synaptic inhibitory synapses. This is now explicitly mentioned page 9 of the revised manuscript: “specifically expressed at the postsynaptic site of all GABA monoaminergic synapses¹⁰”.

As per changes in expression of individual genes in the RNAseq: in response to the comment of reviewer 1, we decided to use the same (stringent) pipeline of analysis as the companion manuscript by Sahadevan and collaborators. As a consequence, this conservative pipeline did not identify individually differentially expressed genes between genotypes.

7. About the increased levels of FUS protein at synaptosomal fractions, authors should include controls quantifying FUS levels from total lysates, including nuclear and cytoplasmic fractions. Is the increase of FUS at synapses simply reflecting higher levels of FUS overall in mutant mice due to auto-regulation? Or is there a specific increase of FUS at the synapsis?

We thank the reviewer for this question, and now included this control in the revised manuscript.

We performed western blotting on the total fractions of these experiments, and observed, as expected, an increase in total FUS levels.

Interestingly however, the relative increase in FUS in total extracts was about 3 times, while we observed almost 10 times increased FUS levels in synaptosomal fractions, thus arguing for a selective synaptic accumulation of FUS in this mutant mouse model. Of note, this synaptic accumulation of mutant FUS is also observed in independent experiments, using different protocols, in the accompanying manuscript of Sahadevan and collaborators.

8. Finally, in the section “Defects in inhibitory neurons in Fus Δ NLS/+ mice” on line number 256 it should refer to Figure 5C, not Figure 5D.

We apologize for this error.

This section was heavily rewritten during revision, and we double checked for such errors throughout the manuscript.

Reviewer #3 (Remarks to the Author):

FUS-ALS, caused by C-terminal FUS mutations leading to FUS cytoplasmic mislocalisation, constitutes the most severe form of ALS, and although rare, is representative of RNA binding protein dysfunction/misregulation very commonly observed in ALS and FTD. Furthermore, FUS mislocalisation has also been observed in non-FUS-mutated cases of ALS, while FUS cytoplasmic mislocalisation is observed in a subset of FTD cases, albeit without mutations in most cases. Taken together, the findings presented in this paper are of relevance to the wider field of ALS/FTD. FUS is known to be a dosage sensitive gene, and while other, transgenic, animal models of FUS-ALS display stronger phenotypes, the knock-in model used in this study provides better physiological relevance due to the endogenous expression levels (and endogenous control of expression) of mutated FUS.

The authors have previously investigated the spinal cord and motor function in this model, and here investigate behavioral phenotypes and brain dysfunction, given the links of FUS to FTD. As a ubiquitously expressed gene, it is important to understand the wider effects (beyond motor neurons) of FUS mislocalisation to fully understand FUS-related disease – and in this respect the authors provide strong overall evidence of an inhibitory cortical neuron synaptopathy.

Behavioral analyses show relevant phenotypes (although I have some reservations and queries) that nicely tie in with cortical synaptic dysfunction - specifically inhibitory synapse function - to provide a mechanistic explanation. Considering the mouse utilized is a physiological model, and that mice have a relatively short lifespan, a lack of neurodegeneration is not too surprising and should not be considered problematic. The mechanisms described provide clues regarding early pathogenic changes critical to understanding and treating FUS-related disease.

Behavioral analyses

The authors ran a series of behavioural tests on male mice. Were female mice not at all tested, or did they not show a phenotype?

We thank the reviewer for this important question and provide details on genders used in all experiments in both materials and methods and figure legends.

Most of the experiments, as correctly noted by the reviewer, have been performed in male mice, in particular all behavioral analysis. Female mice were only included for (i) electron microscopy (in a gender balanced group), (ii) in vivo recordings (in a gender balanced group), and (iii) for synaptosomal preparations.

We fully agree that ideally both sexes should be included for behavioural experiments. However, these experiments were carried out as part of a larger ethical application that included also the study of homozygous Δ NLS mice². In this context, most of the female Δ NLS/+ mice were used for breeding purposes. In order to minimize the numbers of animals used in these experiments, and minimize the numbers of animals generated in the overall project, and thus adhere as strictly as possible to the 3R rules, our animal license only included behavioural analysis in male mice (Cremeas ethical committee reference number AL/27/34/02/13).

In addition, a large number of studies have documented gender differences in the prevalence of FTD. With respect to FTD, Ratnavalli and colleagues⁷ found that men were four times more likely than women to be affected by bvFTD. Johnson and colleagues also reported sex differences in clinical presentation of FTD⁸. While it is currently not known whether there could be substantial differences in FTD-FUS, we think that the investigation of possible gender differences in the phenotypes of *Fus* ^{Δ NLS/+} mice goes beyond the current manuscript and would require the generation of parallel cohorts of male and female mice. However, in several cross

breeding experiments that have been performed in other contexts than the current study, we observed similar defects in female Δ NLS mice, at least in motor behaviors (inverted grid), and we did not observe major differences between male and female mice in the experimental read-outs in which both sexes have been studied (in vivo recordings, electron microscopy).

To take into account this reviewer's comment:

- we have reviewed our manuscript in order to make more explicit in each figure the sex of the mice being investigated.
- It is now explicitly stated in the materials and methods section that we could not study both male and female mice for practical and ethical reasons to minimize the number of mice generated for these studies.
- The discussion section was also modified to explicitly state that possible sex differences might exist and have not been studied here.

Spontaneous locomotor activity was increased, similar to findings from previously reported FUS models. There seems to be some confusion with the statements: "Hyperactivity was not caused by reduced anxiety," and later "hyperactivity did not appear to be associated with increased anxiety". Either way (I think reduced anxiety is more likely to confound hyperactivity), the authors try to rule out anxiety as a confounding factor, but only test this at 10 months, when hyperactivity is not present (no reduced central time in open field, little evidence of daytime hyperactivity in home cage analysis). I therefore feel the authors should not explicitly rule out the possibility of an anxiety phenotype, in particular at the early 4 month timepoint when hyperactivity is most prevalent. If hyperactivity diminishes as the mice age – could the same age-related process not have the same effect on an anxiety effect? Behavioural changes in FTD are complex and include a range of symptoms such as hyperactivity and anxiety, but also apathy and disinhibition. As such I'm not sure why it would be necessary to rule out confounding factors that may in fact be relevant to disease, and may yield from a common underlying synaptic dysfunction. In short, I would not claim to rule out an anxiety phenotype linked to hyperactivity, simply that you saw no evidence for it in aged mice.

We thank the reviewer for this comment and have modified the text accordingly.

Executive dysfunction was assessed using MWM, where I feel claims regarding "poor performance" are overplayed. Mutants find the platform in the same timeframe as WTs, and while they do spend a statistically significant greater amount of time in random quadrants versus the target quadrant, the actual difference in time is marginal. Similarly, the statement "Fus Δ NLS/+ mice extinguished their previous memory much faster than wild type mice" is an overstatement (Fig 1g). There is an effect, but it seems penetrance and the size of effect is very low.

We thank the reviewer for this comment and have toned down the title and description of results of the corresponding results section.

Sociability was measured in 2 distinct tests at 3 time points. The resident intruder paradigm consistently showed - at 4, 10 and 22 months - mutants spending more time interacting with the intruder. The 3-chamber test only showed increased sociability at 10 months, but not at the earlier timepoints. Can the authors speculate on why there may be a discrepancy between tests? Do the authors think increased sociability could be linked to hyperactivity and impulsivity?

We thank the reviewer for this point. We speculate that discrepancy between results of social

interaction at earlier time points arises from the differences among the corresponding tests, since the resident intruder test measures both sociability and aggressive behavior while in the three chambers test the possibility of aggressive behavior is reduced by physical separation of mice and re-location from home cage to novel environment. We think that lack of direct intrusion that is present in resident intruder test may affect in part social interest in three chamber test. Indeed, only mice at 4 months of age show aggressive behavior similar in both genotypes.

Although increased sociability could be linked to hyperactivity it is unlikely the case in our study since mice show hyperactivity during the night period in the familiar environment and behavioral test were performed during the light phase (between 9 am and 5 pm.). Also, in the three chamber paradigm mice of both genotypes were spending same total time in side chambers – exploration time of novel environment (data not shown), while *Fus^{ANLS/+}* mice were constantly and significantly more interacting with their conspecific compared to empty wired cage. These results indicated once again the absence of hyperactivity in a novel environment in accordance with results obtained by open field task and stated in the result part (page 4 and Figure S1a-b) and in contrast to hyperactivity observed in home cage. Together this suggest the specificity of sociability impairment. However, we cannot exclude possibility that altered impulsivity could contribute to increased sociability, and this would require s as for in the present study we did not perform the explicit tests of impulsivity such as the five-choice serial reaction time task (5-CSRT), the stop-signal task (SST), the go/no-go task or others.

Neuronal activity

The authors clearly demonstrate increased neuronal activity in vivo, in the frontal cortex, in 10 month old mice.

MRI

MRI imaging showed ventricular enlargement in 12 month old mutant mice, but was not due to changes in the cortex or neuron loss. Authors state that this could be due to “neuronal dysfunction,” but is there a precedent for this? Could the morphological changes be developmental in origin, since cases of FUS-ALS are often juvenile in nature and developmental abnormalities have been reported in the literature (e.g. developmental delay)? Indeed, at least 2 developmental genes - *Dlx5* and *Dlx6* - were picked up in transcriptomics.

The sentence the reviewer was referring to was misleading: we did not hypothesize that ventriculomegaly was caused by neuronal dysfunction, but we are rather hypothesizing that altered behavior was caused by neuronal dysfunction. This section has been largely rewritten in response to comments of reviewer 1 and we modified this misleading sentence.

Transcriptomics

Frontal cortex RNAseq was performed at 5 and 22 months. Authors describe an intriguing predominance of upregulation of genes at 5 months and downregulation at 22 months – and GABA related genes in particular were first upregulated then downregulated, pointing to inhibitory neuron dysfunction, tying in with neuronal hyperexcitability. Comparison was made to previous datasets, including *FUS*^{-/-} mice. It has been shown that transcriptional changes associated with *FUS* mutation are consistent with loss of function (i.e. changes strongly overlap in the same direction with *FUS*^{-/-} mice/cells). How do the results here chime with previous datasets in terms of directionality of gene changes – i.e. do changes in gene expression here

chime with loss of function at early or late timepoints, or do the authors find novel gain of function changes?

Further to reviewer's 1 comments, we harmonized the analytical pipelines between the two co-submitted manuscripts and performed a number of new bioinformatic analyses. This section has thus been heavily rewritten and the comparison with knock out mice has been removed from the manuscript. We also have discussed these new analyses and their relevance in this new version.

Defects in inhibitory neurons

Inhibitory cortical PV interneurons were further investigated in light of the observed transcriptomic changes, and markers associated with this population were found to be altered, and inhibitory synapse markers were also shown to be altered at the protein level.

Of note, ultrastructural analyses showed dysmorphology of inhibitory cortical synapses, together with the observation that inhibitory synaptic density was reduced. While these were only shown at 22 months, in the accompanying paper inhibitory synapse defects were shown at early timepoints; altogether, suggestive of causation with respect to neuronal hyperexcitability.

Abnormal synaptosomal fractions

Finally, synaptosomal fractions were analysed via qPCR and Western Blotting in 5 month old mice. Evidence for mutant, but not WT, FUS protein was found to be enriched at synapses – as might be expected due to lack of nuclear localization signal on the mutant protein. At the mRNA level, 3 FUS binding mRNAs: Fus, Nrnx1, and Gabta1, were also enriched at the synapse in mutants (the latter 2 involved in inhibitory synapse function). Can the authors please discuss further, here or in the discussion, the potential mechanism of Nrnx1 and Gabta1 mRNA increases (i.e. how could mutant FUS cause this?), and how increases in the mRNAs fit in with their hypothesis of reduced inhibitory synapse function?

Further to this comment, we have introduced a more mechanistic discussion on the relationships between mRNA synaptic increases, altered synaptic physiology and altered behavior.

Can the authors please explain why they chose to look at Chrna7, and what the significance might be?

The reviewer is right in pointing out that this gene had little relevance to the current study. We removed it and performed new RT-qPCRs on hub genes identified in the WGCNA approach.

Discussion

In the sentence, "...we observed an early and sustained spontaneous locomotor hyperactivity in Fus Δ NLS/+ mice" - I don't agree with "sustained", as at 10 months, hyperactivity levels were

far more restricted, and not at all present from the open field test; and the later 22 month time point was not performed.

The sentence was corrected.

As stated above, I think the authors cannot rule out anxiety being linked to hyperactivity.

The sentence stating that anxiety was not involved in these behavioral alterations was removed.

Final comments

In summary, I find the work as a whole to convincingly show that physiological levels of FUS mislocalisation can lead to inhibitory synapse defects and behavioural phenotypes linked to increased cortical activity that may be of important relevance to the wider field of ALS/FTD, and will likely spur further investigation in other mouse models, and patients.

Thomas
MRC Harwell, Oxfordshire, UK

Cunningham

We thank the reviewer for this very positive appreciation of our work.

References

1. Dana, H. *et al.* High-performance calcium sensors for imaging activity in neuronal populations and microcompartments. *Nat Methods* **16**, 649-657 (2019).
2. Scekcic-Zahirovic, J. *et al.* Toxic gain of function from mutant FUS protein is crucial to trigger cell autonomous motor neuron loss. *EMBO J* **35**, 1077-1097 (2016).
3. Scekcic-Zahirovic, J. *et al.* Motor neuron intrinsic and extrinsic mechanisms contribute to the pathogenesis of FUS-associated amyotrophic lateral sclerosis. *Acta Neuropathol* **133**, 887-906 (2017).
4. Picchiarelli, G. *et al.* FUS-mediated regulation of acetylcholine receptor transcription at neuromuscular junctions is compromised in amyotrophic lateral sclerosis. *Nat Neurosci* (2019).
5. Martinez-Silva, M.L. *et al.* Hypoexcitability precedes denervation in the large fast-contracting motor units in two unrelated mouse models of ALS. *Elife* **7**(2018).
6. Gunes, Z.I., Kan, V.W.Y., Ye, X. & Liebscher, S. Exciting Complexity: The Role of Motor Circuit Elements in ALS Pathophysiology. *Front Neurosci* **14**, 573 (2020).
7. Ratnavalli, E., Brayne, C., Dawson, K. & Hodges, J.R. The prevalence of frontotemporal dementia. *Neurology* **58**, 1615-21 (2002).
8. Johnson, J.K. *et al.* Frontotemporal lobar degeneration: demographic characteristics of 353 patients. *Arch Neurol* **62**, 925-30 (2005).

9. Zhang, Y., Parmigiani, G. & Johnson, W.E. ComBat-seq: batch effect adjustment for RNA-seq count data. *NAR Genom Bioinform* **2**, lqaa078 (2020).
10. Fritschy, J.M. & Mohler, H. GABAA-receptor heterogeneity in the adult rat brain: differential regional and cellular distribution of seven major subunits. *J Comp Neurol* **359**, 154-94 (1995).

REVIEWER COMMENTS

Reviewer #1 (Remarks to the Author):

New experiments and descriptions provided in the revised manuscript have significantly strengthened this work — I am particularly impressed by the new analysis of neuronal activity in 4mo old animals, as well as the re-analysis of RNA-seq data, making this manuscript much more consistent with the companion manuscript by Sahadevan et al.

Reviewer #2 (Remarks to the Author):

The manuscript is now acceptable for publication

Reviewer #3 (Remarks to the Author):

In my opinion the authors have done a good job of addressing mine and the other reviewers' comments. I'm satisfied that the conclusions drawn from their data stand, and therefore happy for the manuscript to be published in the current form.